# SMGRL: Scalable Multi-resolution Graph Representation Learning

## Abstract

Graph convolutional networks (GCNs) allow us to learn topologically-aware node embeddings, which can be useful for classification or link prediction. However, they are unable to capture interactions between nodes that are not direct neighbors without adding additional layers— which in turn leads to over-smoothing and increased time and space complexity. Further, the complex dependencies between nodes make mini-batching challenging, limiting their applicability to large graphs. We propose a Scalable Multi-resolution Graph Representation Learning (SMGRL) framework that enables us to learn multi-resolution node embeddings efficiently. Our framework is model-agnostic and can be applied to any existing GCN model. We dramatically reduce training costs by training only on a reduced-dimension coarsening of the original graph, then exploit self-similarity to apply the resulting algorithm at multiple resolutions. The resulting multi-resolution embeddings can be aggregated to yield high-quality node embeddings that capture interactions occurring at multiple lengthscales (i.e., between pairs of nodes with a variety of minimum path lengths). Our experiments show that this leads to improved classification accuracy, without incurring high computational costs.

## 1 Introduction

When working with graph-structured data, we often wish to learn latent vector representations for nodes within the graph—often referred to as node embeddings. These representations, which typically aim to capture the topological structure of the graph, can be used for tasks such as node classification and link prediction, or can be combined to obtain a representation of the entire graph. Message passing algorithms, where a node's embedding is updated based on its neighbors' embeddings, are a natural way to capture topological structure. Graph convolutional networks (GCNs, Kipf & Welling, 2017; Hamilton et al., 2017; Veličković et al., 2018) learn the form of these updates using neural networks.[1] Since GCNs learn an algorithm to obtain embeddings (rather than directly learn a mapping from nodes to embeddings), we can apply the resulting algorithm to new or modified graphs, allowing GCNs to operate in an inductive fashion. This gives them a clear advantage over transductive algorithms such as node2vec (Grover & Leskovec, 2016) or DeepWalk (Perozzi et al., 2014), which learn a one-to-one mapping from node to embedding, often based on random walks over the graph, and cannot generalize to nodes not appearing in the training graph.

GCNs have achieved impressive performance on many graph-based classification, prediction, and simulation tasks (Klicpera et al., 2019; Brockschmidt, 2020; Bianchi et al., 2021). However, there are limits to their representational power. A single-layer GCN aggregates information only from a node's immediate neighborhood, ignoring interactions operating at longer lengthscales.[2] This can be addressed by adding more layers—a $K$-layer GCN incorporates information from a node's $K$-hop neighborhood. For small values of $K$, this can improve representational power; however as the reach of the message passing algorithm expands,

---

[1]The original GCN (Kipf & Welling, 2017) was so-named because it deploys a message-passing algorithm that approximates spectral convolution on the graph; in this work, we use the term more broadly to cover algorithms that learn a message-passing algorithm on a graph.

[2]By the lengthscale of an interaction between two nodes, we refer to the minimum path distance between those two nodes. In the case of a single-layer GCN, all interactions are at a lengthscale of 1.

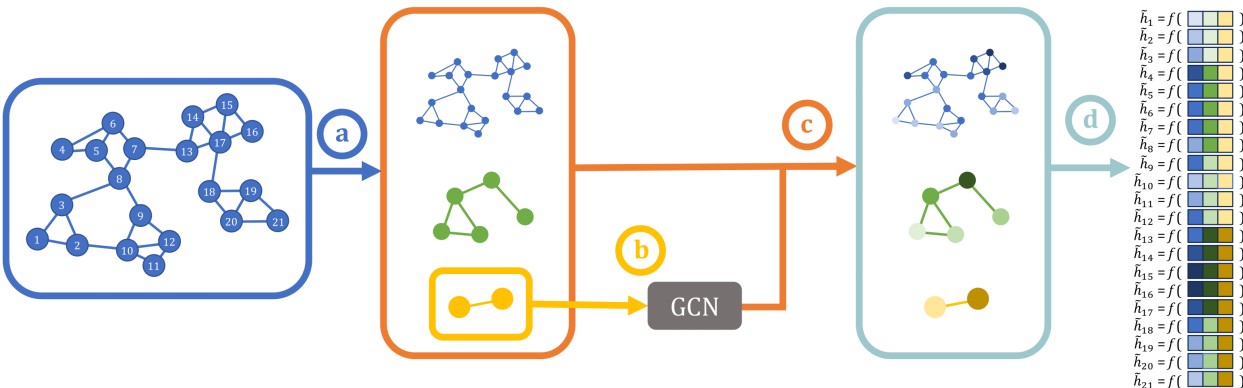

Figure 1: A schematic of SMGRL. a) We coarsen our original graph to obtain a hierarchy $\mathcal{G}_0, \ldots, \mathcal{G}_L$. b) We train our GCN on the coarsest graph, $\mathcal{G}_L$. c) We use the trained GCN to obtain embeddings at each layer of the hierarchy. d) We combine the per-layer embeddings to get an overall representation for each node.

the inferred representations become increasingly homogeneous—a phenomenon known as oversmoothing (Li et al., 2016; Oono & Suzuki, 2019; Cai & Wang, 2020)—leading to a decrease in performance.

In addition, the computational and memory requirements of GCNs scale poorly when compared with standard feedforward neural networks. The number of messages—and hence the computational cost—scales with the number of edges, which can grow quadratically in the numbere of nodes. Memory requirements are inflated due to the connectivity pattern of the graph: the representation of a node relies on all the nodes in its $K$-hop neighborhood. This means that, when minibatching, we must store in memory not just the set of nodes selected for the minibatch, but also their $K$-hop neighborhood (or an appropriately informative subset thereof).

One way to reduce the computational and memory requirements is to train the GCN on a smaller graph that is "similar" to the original graph. Loukas (2019) showed that a class of coarsening algorithms can be used to construct graphs that have similar spectral properties to the original graph. Huang et al. (2021) show that training a GCN on this coarsened graph can be seen as an approximation to training on the full graph, and that we can use the resulting algorithm on the original graph to obtain node embeddings. Moreover, theoretical and empirical results suggest that this procedure effectively acts as a regularizer, leading to improved generalization performance over the baseline GCN.

Since the GCN deployed on the coarsened graph is ultimately deployed on the original graph to obtain node embeddings, the approach of Huang et al. (2021) does not address the question of how to capture interactions occurring at longer lengthscales without increasing the number of layers in the GCN (increasing computational cost and risking washing out the impact of local structure due to oversmoothing). Instead, we construct a hierarchy of graphs $\mathcal{G}_0, \ldots, \mathcal{G}_L$ by aggregating nodes at level $\ell$ into "supernodes" at level $\ell + 1$. The intuition here is that one-hop neighborhoods in the coarser-resolution graphs at the top of the hierarchy capture longer lengthscale behavior such as interactions between communities and cliques, while one-hop neighborhoods in the finer-resolution graphs at the bottom of the hierarchy capture local interactions. Rather than learn embeddings separately for each level of the hierarchy, we infer level-specific embeddings using a single GCN trained on the coarsest level. The resulting embeddings—each capturing information at a different lengthscale—can be combined into a single, multi-resolution embedding for each node. This approach, which we summarize in Figure 1, is reminiscent of hierarchical methods that learn representations at multiple levels of a hierarchy (Chen et al., 2018; Akyildiz et al., 2020; Liang et al., 2021; Zhong et al., 2022; Guo et al., 2021; Jiang et al., 2020). However, unlike such methods, we do not incur high computational costs associated with training or refining at each level of the hierarchy.

We find that the resulting algorithm—whose computational and memory requirements are almost identical to those of the coarsened GCN approach of Huang et al. (2021)—typically outperforms both a GCN trained

on the original graph, *and* the coarsened GCN algorithm, which only considers a single resolution. It also outperforms several existing hierarchical GCN algorithms, while having a much lower computational cost.

Our framework, which we denote Scalable, Multi-resolution Graph Representation Learning (SMGRL), consists of three components: a hierarchical graph coarsening algorithm (step a in Figure 1); a GCN learned on a coarsened graph (step b) and deployed at all levels of the hierarchy (step c); and a final aggregation step (step d). Each of these components is highly customizable. In particular, we explore three choices of GCN, and in the appendix explore the impact of different aggregation schemes.

## 2 Background

### 2.1 Graph convolutional networks

Representing nodes $v_i \in \mathcal{V}$ within a graph $\mathcal{G} = (\mathcal{V}, \mathcal{E})$ in terms of embeddings $h_i \in \mathbb{R}^d$ allows us to distill important information about the graph topology in an easily digestible package. GCNs allow us to learn such embeddings in an inductive manner: rather than directly learn a mapping from $\mathcal{V}$ to $\mathbb{R}^d$, GCNs learn how to learn such embeddings. Concretely, they parametrize a message-passing algorithm that allows us to update node embeddings by passing messages along the graph's edges, and aggregating the messages arriving at each node.

Consider a node classification task where each node $v_i$ has feature $x_i$ and label $y_i$. Our goal is to find node embeddings $h_i$, and a corresponding classification function $g_\theta$, that minimize some loss $\sum_{i=1}^{|\mathcal{V}|} \mathcal{L}(g_\theta(h_i), y_i)$. We do so by learning a message passing update rule of the form

$$h_i \leftarrow \text{UPDATE}\left(x_i, \text{AGGREGATE}\left\{x_j : v_j \in \text{Ne}(v_i; \mathcal{G})\right\}\right), \tag{1}$$

where $\text{Ne}(v_i; \mathcal{G})$ indicates the neighborhood of $v_i$ in $\mathcal{G}$. Multiple layers of the form in (1) can be stacked so that the embedding is based on a multi-hop neighborhood.

When $\text{UPDATE}(m)$ takes the form $\sigma(Wm)$ and $\text{AGGREGATE}$ is an appropriately normalized summation, this can be seen as an approximation to graph convolution Kipf & Welling (2017). In this paper, we use the term graph convolutional network more broadly to refer to any message passing algorithm of the form of (1), allowing for more expressive aggregation methods (e.g., Xu et al., 2018; Veličković et al., 2018). We also include methods that use alternative definitions of neighborhoods Hamilton et al. (2017); Klicpera et al. (2019). By contrast, we use the terminology "graph neural network" to also include non-message-passing node and graph representation algorithms such as node2vec (Grover & Leskovec, 2016).

GCNs have proved a powerful tool for learning node embeddings. However, they do have a number of limitations. Firstly, GCNs typically have high time and space complexity. Training a GCN involves updating each node's embedding based on it's neighborhood once for each layer, meaning that the computational complexity of training a $K$-layer GCN scales as $O(K|\mathcal{E}|)$. If our training graph is dense, $|\mathcal{E}| \sim O(|\mathcal{V}|^2)$. Further, dependency between nodes makes minibatching challenging: if a GCN has $K$ layers, then updating the embedding of a single node requires knowledge of the $K$-hop neighborhood of that node. This leads to increased memory requirements when minibatching, since we must augment a size-$m$ minibatch with its $K$-hop neighborhood. In dense graphs, this can easily envelop a high proportion of the full graph.

Secondly, in a single-layer GCN the embedding of a node depends only on the node features of its immediate neighborhood. This limits our ability to learn relationships occurring at longer lengthscales, for example dependencies on nodes more than a single edge away. To get around this, we can stack multiple layers, increasing the sphere of influence. However, adding more layers leads to over-smoothing—all embeddings become increasingly similar—which in turn leads to degradation in prediction performance (Li et al., 2016; Oono & Suzuki, 2019; Cai & Wang, 2020).

### 2.2 Graph coarsening and hierarchical embeddings

In many cases, graphs can be thought of as the realization of a latent hierarchical structure. For example, in a road network, we can think of a top-level graph connecting cities via major thoroughways, and a

lower-level graph that also includes local roads. In a social network, we can think of individuals belonging to communities (such as schools, workplaces or hobbies), and consider the degree of connectivity between communities as well as the pattern of connectivity within communities. Such hierarchical representations allow us to look at the graph at different resolutions, or lengthscales: the original graph allows us to inspect local interactions (occurring on the lengthscale of one degree of separation between individuals), while the coarser representations allow us to consider interactions occuring at longer lengthscales (i.e., lengthscales corresponding to much higher average degrees of separation between individuals).

In general, an explicit hierarchy is not provided; however we can use various graph coarsening, node clustering or community detection algorithms to infer a hierarchy (e.g., Karypis & Kumar, 1998; Blondel et al., 2008; Loukas, 2019; Cai et al., 2021). Coarsening a graph involves constructing a sequence of graphs $\mathcal{G}_0, \ldots, \mathcal{G}_L$, such that $\mathcal{G}_0 := \mathcal{G}$ and nodes in $\mathcal{G}_\ell$ have a single parent in $\mathcal{G}_{\ell+1}$. We refer to the original graph $\mathcal{G}_0$ as the bottom layer of the hierarchy, and $\mathcal{G}_L$ as the top layer.

By learning embeddings at multiple layers in the hierarchy, we can capture variation at multiple levels of resolution. For example, we might jointly train GCNs at multiple levels in a hierarchy (Jiang et al., 2020; Guo et al., 2021), or learn a single GCN that spans all levels (Zhong et al., 2022). Such approaches have been found to increase expressive power over GCNs learned on the original graph. In addition, hierarchies have been successfully exploited to learn informative representations of entire graphs. For example, hierarchical pooling methods have been used to combine representations from multiple levels in a hierarchy, resulting in a multi-resolution graph embedding (Ying et al., 2018; Huang et al., 2019; Bandyopadhyay et al., 2020; Xin et al., 2021; Liu et al., 2021).

A related family of approaches use a hierarchical representation to refine embeddings (typically learned in a transductive manner). For example, we might learn embeddings at the coarsest layer $\mathcal{G}_L$, and then sequentially refine them to get embeddings at finer levels (Chen et al., 2018; Akyildiz et al., 2020; Liang et al., 2021). Or, we might start by learning embeddings on the original graph $\mathcal{G}_0$, coarsen these embeddings by propagating them up the hierarchy, and then refine them by propagating down the hierarchy (Hu et al., 2019; Li et al., 2020). While these approaches do not explicitly use multi-level representations in the final embedding, the refinement process biases the representations towards including information from the coarsened graphs.

An alternative use of graph coarsening is to reduce the computational cost of training a GCN. If our coarsened graph $\mathcal{G}_L$ is structurally similar to the original graph $\mathcal{G}_0$, we can train our GCN on $\mathcal{G}_L$ and then deploy it directly on $\mathcal{G}_0$. Coarsening algorithms that aim to maintain spectral properties of the original graph are an appropriate choice here (Loukas & Vandergheynst, 2018; Loukas, 2019; Bravo Hermsdorff & Gunderson, 2019; Cai et al., 2021). These representations are well-matched to the task of learning GCNs, which (for certain choices of UPDATE and AGGREGATE in Equation 1) can be viewed as generalizations of approximate spectral convolution over the graph (Kipf & Welling, 2017). Indeed, Huang et al. (2021) show that a GCN trained on a spectrally coarsened graph can be seen as an approximation to a GCN trained on the full graph. In addition to reducing computation and memory requirements, training a GCN on a spectrally coarsened graph often leads to improved generalization, likely due to the approximation mechanism acting as a regularizer.

## 3   The proposed framework: SMGRL

Often, the label of a node in a graph is related not only to the labels of its immediate neighbors and the structure of its immediate neighborhood (relationships we describe as occurring at short lengthscales), but also to the labels of nodes further away in the graph and larger-scale structural properties of the graph (relationships we describe as occurring at short lengthscales. Hierarchical node embeddings, such as those discussed in Section 2.2, allow us to aggregate information at multiple resolutions, incorporating relationships occurring at multiple lengthscales. To avoid this, we exploit the fact that graph coarsening algorithms such as those proposed by Loukas (2019) are explicitly designed to preserve spectral properties of the original graph. This suggests that a message passing algorithm learned at one level of granularity might be appropriate at multiple levels. This intuition is made concrete by the coarsened GCN framework of Huang et al. (2021), who train on the coarsest level of a spectrally-coarsened hierarchy, and use the resulting algorithm on the original graph.

We go further: we deploy a GCN learned on the coarsest graph at *all* levels of the hierarchy. This leads to a minimal increase in computational cost over Huang et al. (2021), but a potentially significant increase in representational power. If a GCN trained on level $L$ is a good approximation to one trained on level 0, then it should also be a good approximation at levels $0 < \ell < L$. The embeddings obtained at levels $\ell > 0$ will capture variation at longer length scales, without needing to add layers to our GCN. In particular, if the hierarchy obtained via the spectral coarsening algorithm aligns with intuitive notions of interconnected communities within the graph, embeddings at coarser resolutions will capture interactions between such communities.

This approach is reminiscent of the hierarchical GCNs described in Section 2.2, which combine embeddings at multiple resolutions. However, by reusing a GCN learned at the coarsest level, our computational costs are significantly lower than approaches that learn GCNs at all levels. Further, since all embeddings are obtained using a common aggregation rule, they can be linearly combined to obtain a single embedding for each node.

The computational cost of SMGRL scales similarly to that of the coarsened GCN algorithm of Huang et al. (2021); we provide a complexity analysis in Appendix B. The hierarchical coarsening can be pre-computed on CPU; further, since the algorithm is deterministic, we only need compute it once if we are using multiple random seeds or exploring multiple hyperparameter settings for our GCN.

Our framework consists of four steps: a) deploying a hierarchical coarsening algorithm; b) training a GCN on the coarsest graph; c) deploying this GCN on all levels of the hierarchy; and d) aggregating the resulting embeddings. This procedure is visualized in Figure 1.We provide details of each step below, and summarize the framework in Algorithm 1. SMGRL makes a number of assumptions on the structure of $\mathcal{G}$: We assume $\mathcal{G}_\ell$ is undirected and without edge labels, both requirements of our coarsening algorithm. Further, while not explicitly required by the algorithm, we do not suggest the use of SMGRL on highly heterophilic graphs since the coarsening algorithm does not take into account node labels; we discuss this further in Section A.

**a) Construct a coarsened representation** We begin by constructing hierarchically coarsened representations $\mathcal{G}_0, \ldots, \mathcal{G}_L$ of our graph (where $\mathcal{G}_\ell = (\mathcal{V}_\ell, \mathcal{E}_\ell)$ and $\mathcal{G}_0 := \mathcal{G}$), alongside associated coarsened features $X_\ell$ and labels $Y_\ell$. Our goal is to obtain a hierarchy such that a GCN trained on one graph is appropriate for all graphs in the hierarchy. A natural choice is to use a graph coarsening method that (approximately) preserves spectral properties of the original graph.

Loukas (2019) proposes such a method, that constructs coarsening matrices $P_\ell$ such that $L_\ell = P_\ell^\mp L_{\ell-1} P_\ell^+$, where $L_\ell$ is the Laplacian of $\mathcal{G}_\ell$, and $+$ and $\mp$ indicate the pseudoinverse and transposed pseudoinverse, respectively. Each row of the coarsening matrix selects a set of nodes in $\mathcal{G}_{\ell-1}$ to be collapsed into a single node in $\mathcal{G}_\ell$. These sets are chosen greedily from a set of candidate sets $\mathcal{C}$, in order to minimize a spectral distance between the coarsened graph $\mathcal{G}_\ell$ and the previous level $\mathcal{G}_{\ell-1}$,

$$\text{cost}(\mathcal{C}) = \frac{\Pi_{\mathcal{C}}^\perp A_{\ell-1}}{|\mathcal{C}| - 1}, \tag{2}$$

where $\Pi_{\mathcal{C}}^\perp = I - P_{\mathcal{C}}^+ P_{\mathcal{C}}$ and $A_{\ell-1}$ is the target Laplacian for $\mathcal{G}_{\ell-1}$. A number of choices are available to select the candidate sets; unless otherwise stated we follow the "variation neighborhoods" method, where candidate sets are given by a node and its immediate neighbors. See Algorithm 2 in the Appendix for further details.

While the above approach offers guarantees on the spectral similarity of the coarsened graph to the original graph, in practice we find that performance is similar using alternative graph coarsening algorithms. We explore the use of alternative coarsening algorithms in Appendix A.

Once we have obtained our coarsening matrix $P_\ell$ and corresponding coarsened graph $\mathcal{G}_\ell$, we can use $P_\ell$ to propagate features and labels up the hierarchy. For node features (which are assumed fully observed), we have $X_\ell = P_\ell X_{\ell-1}$, following Loukas (2019). Since labels are not fully observed, we propagate labels as the weighted average of the observed children in $\mathcal{G}_{\ell-1}$, i.e. $y_{i,\ell} = \frac{p_{i,\ell} Y_{\ell-1}}{p_{i,\ell} M_{\ell-1}}$, where $p_{i,\ell}$ is the $i$th row of $P_\ell$ and $M_{\ell-1}$ is a binary array indicating the missingness pattern in $Y_{\ell-1}$

**b) Train a GCN on $\mathcal{G}_L$**   We use the coarsest graph $\mathcal{G}_L$, and the corresponding aggregated features $X_L$ and labels $Y_L$, to learn a message passing algorithm $f_\phi$ and a classification function $g_\theta$, as described in Section 2.1. We note that any GCN can be used here.

**c) Use the trained GCN to obtain embeddings at each level of the hierarchy**   Once training is complete, discard the classification function $g_\theta$ and use $f_\phi$ to learn embeddings $H_\ell \in \mathbb{R}^{|\mathcal{V}_\ell \times d|}$ at each level $\ell$ in the hierarchy. We use our sequence of projection matrices $P_\ell$ to lift these embeddings to $H'_\ell \in \mathbb{R}^{|\mathcal{V}| \times d}$ as $H'_\ell = P_1^+ \cdots P_{\ell+1}^+ H_\ell$, where $P_\ell^+[i,j] = 1$ iff $P_\ell[i,j] > 0$. This differs from Huang et al. (2021), who only learn embeddings for the original graph $\mathcal{G}$.

**d) Aggregate the embeddings and learn a new classification function**   Each node $v_i \in \mathcal{V}$ is now associated with a sequence $h'_{0,i}, \ldots, h'_{L,i}$ of embeddings. We can aggregate these embeddings to obtain a final representation $\widetilde{h}_i = \text{COMBINE}(h'_{0,i}, \ldots, h'_{L,i})$ for each node $v_i$, and learn a new classification function $g_{\widetilde{\theta}}$ based on this new embedding. A number of choices can be made here. A lightweight option is to either concatenate the embeddings, or take a simple mean. Alternatively, if computational resources allow, we can learn a weighted average alongside the classification function, i.e. $\widetilde{h}_i = \sum_{\ell=0}^{L} w_\ell^T h'_{\ell,i}$. Unless otherwise stated, we use this weighted average approach; we explore alternatives in Appendix E.

## 3.1   Applicability of SMGRL

Like the graph coarsening algorithms used in this work, we assume that our graphs are undirected and do not include edge labels. Further, the nature of the graph coarsening algorithms used suggest that our method is primarily appropriate for homophilic graphs. The coarsening algorithms used on this paper look only at the topology of the graph; they do not take into consideration node labels. This is not a problem in homophilic graphs, where neighbors tend to have similar labels, meaning the nodes in each coarsened graphs are likely to correspond to a set of original nodes with the same label. However, in heterophilic graphs, nodes in a coarsened graph are likely to combine nodes with different labels in the original graph. As a result, the labels of the coarsened graphs will become increasingly non-discriminative, limiting our ability to learn.

In general, we find that SMGRL is well-suited to graphs with clear clustering structure, which can be exploited by our choice of coarsening algorithm. We discuss this further in Appendix A, where we explore the coarsened graphs obtained on various datasets.

---

**Algorithm 1** SMGRL

**input**  Graph $\mathcal{G} = (\mathcal{V}, \mathcal{E})$, features $X$, labels $Y$, hierarchical coarsening algorithm COARSE (e.g., Algorithm 2), embedding aggregation method COMBINE, GCN $f_\phi$, classification function $g_\theta$.

**output**  Node embeddings $\widetilde{h}_i$, trained GCN $f_\phi$, trained classification function $g_{\widetilde{\theta}}$

1: $\{\mathcal{G}_\ell, P_\ell\} = \text{COARSE}(\mathcal{G})$
2: **for** $\ell = 1, \ldots, L$ **do**
      : $X_\ell = P_\ell X_{\ell-1}$ **for** $i = 1, \ldots, |\mathcal{V}_i|$ **do**
3:       $y_{i,\ell} = \frac{p_{i,\ell} Y_{\ell-1}}{p_{i,\ell} M_{\ell-1}}$, where $M_{\ell-1}$ is the missingness pattern for $Y_{\ell-1}$
6:    **end for**
7: **end for**
8: Learn parameters $\phi$ and $\theta$ to minimize $\sum_{i=1}^{|\mathcal{V}_L|} \mathcal{L}\left(g_\theta\left(f_\phi(x_{L,i}, \{x_{L,j} : v_{L,j} \in \text{Ne}(v_i)\})\right)\right)$
9: **for** $\ell = 0, \ldots, L$ **do**
10:    Obtain embeddings $h_{\ell,i}$ by applying $f_\phi$ to $\mathcal{G}_\ell$.
11: **end for**
12: Propogate embeddings such that $H'_\ell = P_1^+ \cdots P_{\ell+1}^+ H_\ell$
13: Learn parameters $\widetilde{\theta}$ (and if appropriate, parameters of COMBINE($\cdot$)) to minimize $\sum_{i=1,\ldots,|\mathcal{V}_0|} \mathcal{L}\left(g_{\widetilde{\theta}}(\text{AGG}(h'_{0,i}, \ldots, h'_{L,i}), y_i)\right)$
14: $\widetilde{H}_i \leftarrow \text{COMBINE}(H'_0, \ldots, H'_L)$

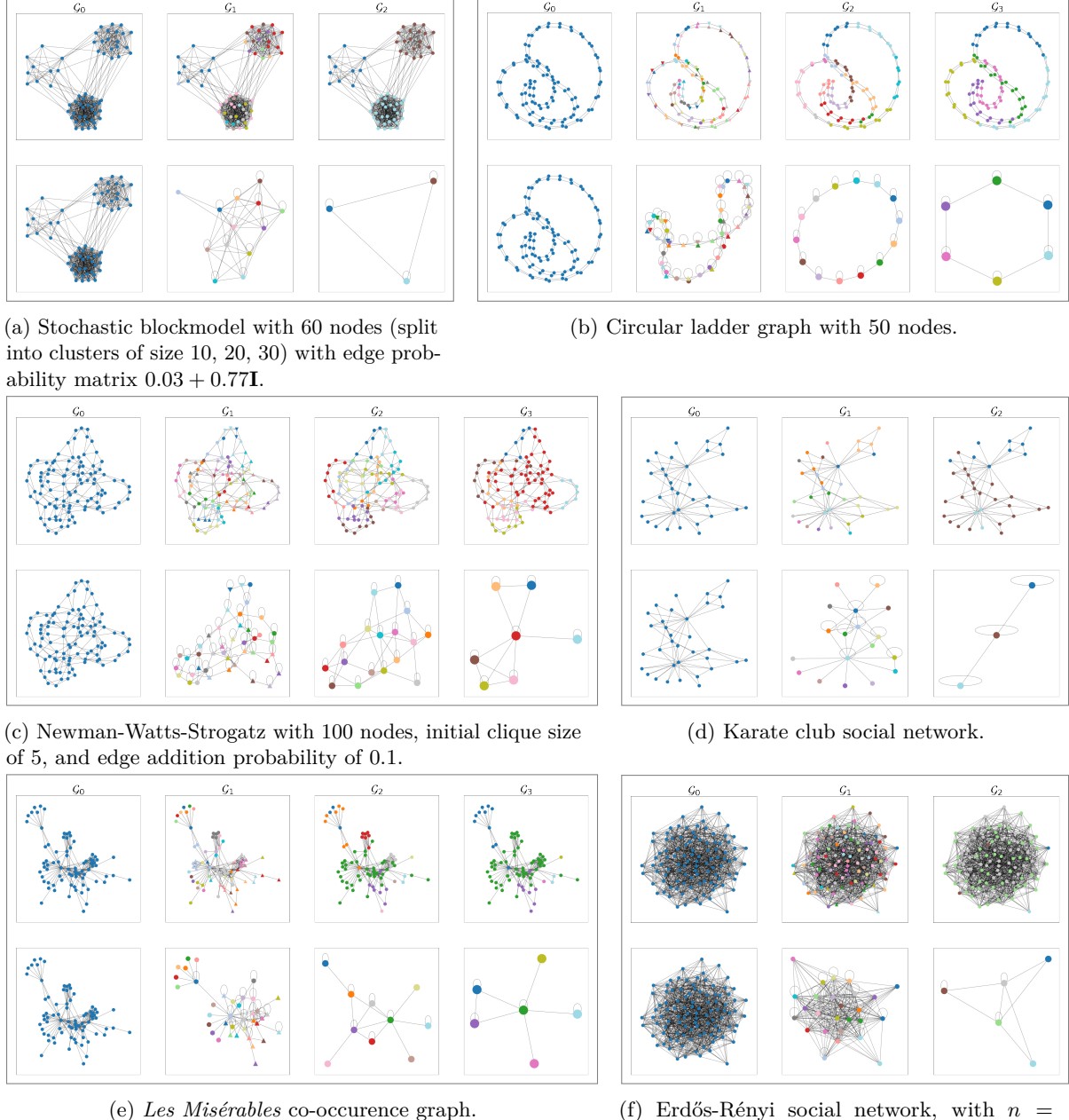

(a) Stochastic blockmodel with 60 nodes (split into clusters of size 10, 20, 30) with edge probability matrix $0.03 + 0.77\mathbf{I}$.

(b) Circular ladder graph with 50 nodes.

(c) Newman-Watts-Strogatz with 100 nodes, initial clique size of 5, and edge addition probability of 0.1.

(d) Karate club social network.

(e) *Les Misérables* co-occurence graph.

(f) Erdős-Rényi social network, with $n = 100, p = 0.2$.

Figure 2: Graph hierarchies obtained on various graphs, using a coarsening ratio of 0.95. Top row: Original graph $\mathcal{G}_0$, with node colors corresponding to the parent node in $\mathcal{G}_n$. Bottom row: Coarsened graph $\mathcal{G}_n$. Colors and shapes are consistent within columns of each subfigure: For example, the red circular nodes in the top row of column $\mathcal{G}_1$ correspond to the red circular node in the bottom row of column $\mathcal{G}_1$. Colors are not consistent between columns.

# 4 Relationship to other hierarchical node embedding methods

SMGRL uses a hierarchical representation of a graph to learn multi-level embeddings, by first training a GCN on the coarsest level of the hierarchy, and then deploying that GCN at all levels. This approach falls under the general family of hierarchical node embeddings, as described in Section 2.2, and shares similarities with several other hierarchical or coarsening-based approaches. In this section, we discuss the most similar methods in detail, and highlight how SMGRL differs from these approaches.

SMGRL makes use of the spectral coarsening algorithm of Loukas (2019). This algorithm was also used by Huang et al. (2021), who train a GCN on the coarsened graph and deploy it on the original graph. Unlike SMGRL, this approach does not incorporate hierarchical information: the coarsened graph is only used to reduce the computational complexity of training, and the node embeddings obtained on the coarsened graph are discarded. As we see in Section 5, this reduces the expressivity of the representations by ignoring multi-scale relationships.

While Huang et al. (2021) only uses a hierarchical coarsening for computational reasons, other node embedding approaches do incorporate multi-level representations, either by explicitly aggregating representations learned at multiple resolutions or by sequentially refining representations across multiple levels of the hierarchy, as described in Section 2.2. A key advantage of SMGRL is the low computational cost of training: we train a single GCN on a single graph, and then deploy it at multiple levels. This is in contrast to methods such as GOSH (Akyildiz et al., 2020), MILE (Liang et al., 2021) or HARP (Chen et al., 2018), which iteratively refine embeddings as we descend the hierarchy, or methods that jointly learn separate GCNs across the entire hierarchy (Guo et al., 2021; Jiang et al., 2020).

Of the alternative hierarchical node embeddings, MILE is perhaps most similar to SMGRL. Like SMGRL, it begins by learning a hierarchical coarsening, and learning embeddings on the coarsened graphs. However, these embeddings are learned using a transductive method such as node2vec, meaning the trained model cannot be reused at different levels of the hierarchy. Instead, they learn a GCN to refine the embeddings at each level of the hierarchy: nodes are initialized by projecting the previous level's final embeddings, and updated using a GCN. This GCN is shared among all levels, as in SMGRL; however, unlike our approach, MILE does not choose a coarsening designed to encourage the GCN to be generally applicable. Finally, MILE only uses the refined embeddings corresponding to the original graphs, discarding the intermediate embeddings; conversely, we combine embeddings from each level of resolution, making sure we retain information from coarser granularities.

# 5 Experiments

SMGLR is designed to improve upon existing GCN-based node representation-learning algorithms by learning and aggregating multi-resolution embeddings, without incurring excessive computational costs. In this section, we provide empirical evidence that our hierarchical representation increases expressivity (Sections 5.2 and 5.3). We show that training a single GCN and applying it at multiple resolution yields comparable performance to training layer-specific GCNs, while reducing computational cost and facilitating simple aggregation schemes. Indeed, we show that SMGLR outperforms several recent alternative hierarchical embedding methods (Section 5.5). We also show that SMGLR can be used in an inductive setting (Sec 5.6) and explore a distributed variant for settings where inference on the full graph is not feasible (Section 5.7).

## 5.1 Datasets and implementation details

We consider seven multi-label classification datasets, summarized in Table 1. The first five datasets are those used by Huang et al. (2021), our closest comparison method. The last two datasets are much larger graphs from the Open Graph Benchmark dataset (Hu et al., 2020). To assess classification performance on these datasets, we look at the macro $F_1$ score on test-set labels; considering alternative metrics such as accuracy showed similar trends. For a subset of experiments, we include accuracy results in the appendix. Unless otherwise stated, we use the default train/validation/test split associated with each dataset for evaluating the macro $F_1$ score.

Table 1: Dataset overview

| Name | Nodes | Edges | Classes | Features |
|---|---|---|---|---|
| Cora | 2,708 | 10,556 | 7 | 1,433 |
| CiteSeer | 3,327 | 9,104 | 6 | 3,703 |
| PubMed | 19,717 | 88,648 | 3 | 500 |
| DBLP | 17,716 | 105,734 | 4 | 1,639 |
| Coauthor Physics | 34,493 | 495,924 | 5 | 8,415 |
| ogbn-arxiv | 169,343 | 1,166,243 | 40 | 128 |
| ogbn-products | 2,449,029 | 61,859,140 | 47 | 100 |

We apply SMGRL to three popular GCNs: GraphSAGE Hamilton et al. (2017), a GCN that uses skip connections to construct a generalized neighborhood aggregation scheme; APPNP (Klicpera et al., 2019), a GCN that uses an aggregation scheme inspired by personalized PageRank; and self-supervised graph attention networks (SuperGAT, Kim & Oh, 2021), a GCN that incorporates attention-based aggregation and uses self-supervised learning in training. These three methods are designed to show that SMGRL is applicable across a wide range of GCNs, incorporating different notions of neighborhood and different message aggregation schemes. We use existing Pytorch implementations of GraphSAGE, APPNP and SuperGAT[3]. To obtain class predictions from embeddings, we use a single-layer perceptron neural network.

We construct our hierarchy using the variation neighborhoods coarsening method (Loukas, 2019), using the original author's implementation[4]. This approach automatically determines the appropriate hierarchical depth, based on the desired reduction ratio. Unless otherwise stated, we learn per-dimension weights for each embedding to aggregate the multi-resolution embeddings. These weights are learned jointly with our final classification algorithm. In Appendix E, we explore some alternative aggregation schemes.

For both GraphSAGE and APPNP, we use additive aggregation of messages; in SuperGAT, an attention mechanism is used for message aggregation. For APPNP, we use parameters $k = 3$ and $\alpha = 0.5$; for all other methods including HARP and MILE, we use the default parameters suggested by the authors. For all three GCNs, we optimize using RMSProp with default hyperparameters. We explored alternative choices of optimzer and hyperparameters and found the outcomes did not change significantly. We carry out early stopping if validation loss does not improve in twenty epochs, and evaluate using the checkpoint with the best validation loss. For Cora, CiteSeer, PubMed, DBLP and Coauthor Physics, results are averaged over 200 runs with random seeds. For the larger OGBN-arXiv and OGBN-Products datasets, results are averaged over three runs with random seeds. Code to reproduce our experiments is available at `https://anonymous.4open.science/r/SMGRL`.

## 5.2 Incorporating representations at multiple resolutions leads to better embeddings

A single-layer GCN is only able to capture information about its immediate neighbors, which limits our ability to capture interactions occuring at longer lengthscales. This is also true in the coarsened GCN approach of Huang et al. (2021); while a coarsened graph is used to *train* a GCN, the final embeddings are only obtained based on the full graph, so if we use a single-layer GCN we can only aggregate information from a direct neighborhood.

A key motivation behind SMGRL is that incorporating embeddings learned at multiple scales will lead to more informative representations and better performance on downstream tasks. Recall that our argument for improved embeddings relies on the idea that different layers of the hierarchy will capture relationships occurring at different lengthscales. We begin by showing that edges in the coarsened graphs do indeed correspond to different lengthscales. Table 2 shows the average minimum path length in the original graph $\mathcal{G}_0$ between pairs of nodes whose parents are neighbors in the coarsened graph $\mathcal{G}_1$, using a 2-layer hierarchy with coarsening ratio of 0.4. This gives the average lengthscale of interactions when we apply a single-layer GCN to

---

[3]`https://pytorch-geometric.readthedocs.io`
[4]`https://github.com/loukasa/graph-coarsening`

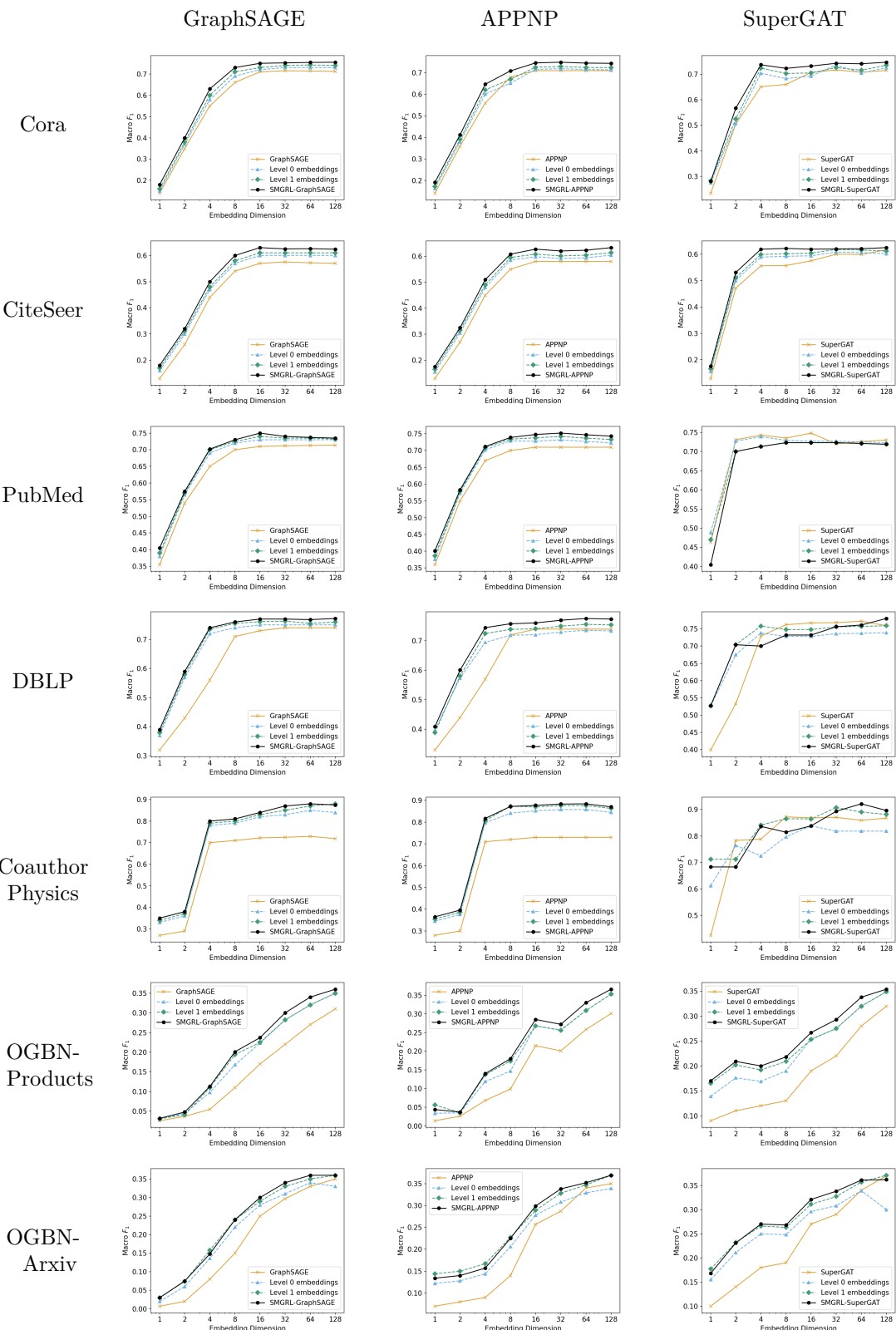

Figure 3: Test-set macro $F_1$ score for SMLRG embeddings obtained at different levels of the hierarchy, using three single-layer GCN architectures (GraphSAGE, APPNP, SuperGAT), a reduction ratio of 0.4 (resulting in a two-layer hierarchy), and various embedding dimensions. Note, the results for Level 0 embeddings directly correspond to the method of Huang et al. (2021).

Table 2: Average shortest path length between pairs of nodes in $\mathcal{G}_0$ whose parents are neighbors in the coarsened graph $\mathcal{G}_1$. This corresponds to the average lengthscale of interactions captured using a GCN on $\mathcal{G}_1$.

| Cora | Cuteseer | Pubmed | Coauthor Physics | DBLP |
|------|----------|--------|------------------|------|
| 2.259 | 2.480 | 2.893 | 2.717 | 2.518 |

$\mathcal{G}_1$ (note that a single-layer GCN on $\mathcal{G}_1$ operates on a lengthscale of 1). We see that the average lengthscale of interactions on the coarsened graph using a single-layer GCN is between 2 and 3— the embeddings obtained using $\mathcal{G}_1$ are indeed able to capture relationships with a longer lengthscale than embeddings ontained using $\mathcal{G}_0$.

To explore the benefits of learning embeddings using different lengthscales, we begin by looking at the embeddings associated with each level in the hierarchy. Figure 3 shows test set macro $F_1$ score across four datasets, using a reduction ratio of 0.4, and with a variety of embedding dimensions (analogous accuracy results are shown in Figure 10 in Appendix C). Each column corresponds to one of the three base GCN algorithms, and each row to a dataset. Within each figure, the dashed lines correspond to embeddings associated with the coarsened graph (level 1) and the original graph (level 0). Recall that both of these embeddings are obtained using the base GCN trained on the coarsened (level 1) graph. The level 0 embedding is therefore equivalent to the graph coarsened approach of Huang et al. (2021). The solid black line corresponds to the SMGRL embedding obtained using a weighted average of the level 0 and level 1 embeddings. In addition, the solid orange line corresponds to embeddings obtained using the base GCN trained on the full graph. In Appendix C, we show comparable results holding embeddings dimension fixed and adjusting the reduction ratio (Figure 9).

We see several interesting patterns here. First, we note that learning either GraphSAGE or APPNP on the coarsened graph and then deploying it on the full graph (level 0) typically outperforms the corresponding baseline GCN. This supports the findings of Huang et al. (2021), who hypothesize that the coarsened graph acts as a regularizer, due to limiting the space of possible convolution operators. Performance is more mixed in the case of SuperGAT: while the level 0 embeddings outperform the full SuperGAT implementation for CiteSeer, OGBN-Products and OGBN-Arxiv, the full SuperGAT performs as well as or better than the level 0 embeddings on the other datasets. This may be because the attention-based aggregation scheme used in SuperGAT is quite distinct from the formulation in Kipf & Welling (2017) (which corresponds to approximate spectral convolution), suggesting that the algorithm might no longer be approximating spectral convolution. If this is the case, then there is less of a clear motivation to train on a spectrally coarsened graph.

Interestingly, in almost all cases where the level-0 embeddings outperform the full GCN, we find that using just the top-level embeddings—i.e., assigning to each node in $\mathcal{G}$ the embeddings obtained for its ancestor in the coarsened graph—actually performs *better* than the embeddings obtained using the same algorithm on the original graph. We hypothesize that this is because the lengthscale associated with the coarser graph may be better aligned with the natural variation in labels.

While the individual embeddings obtain good performance, we find that the full SMGRL embedding (that combines the embeddings from multiple layers using learned weights) tends to lead to better predictive performance than any single layer (although again, performance is more mixed for SuperGAT). In other words: a multi-resolution approach yields improved performance over a model deployed on the original graph, whether that model is trained on the original graph or a coarsened representation (as in Huang et al. (2021). This is not surprising: we are able to aggregate information across multiple lengthscales. In principle, if a given level of the hierarchy does not contain relevant information, we can learn to downweight embeddings from that level in the final aggregation (although a learned aggregation is not guaranteed to outperform any single layer on the test set; for example in the SuperGAT experiments we see several examples where a single layer outperforms the aggregation).

In Appendix C, we repeat this analysis using two-layer GraphSAGE, APPNP and SuperGAT. In general, we see better performance with an additional layer (albeit at higher computational cost), due to the ability to capture relationships occurring at longer lengthscales (a two-layer model can capture interactions at

lengthscales of both one and two edges). As a result, SMGRL is no longer the clear winner in terms of performance, with the unmodified GCNs performing better in many (but not all) cases. However, the faster SMLRG method remains competitive with the full model, and as before, performs better than single-layer embeddings including the approach of Huang et al. (2021).

To summarize: SMGRL provides a lower cost alternative to full GCN training and inference. When the full GCN model is able to capture variation at all appropriate lengthscales, SMGRL provides comparable performance. When the full GCN lacks this ability, SMGRL outperforms the full GCN. Moreover, SMGRL in this scenario SMGRL outperforms the coarsening approach of Huang et al. (2021), for approximately the same computational cost.

### 5.3  SMGRL can capture variation occurring at longer lengthscales

Above, we hypothesized that, when used with a single-layer GCN, SMGRL embeddings are able to perform better than embeddings obtained on the original graph because they are able to capture variation at multiple degrees of resolution. Rather than just propagating information between neighboring nodes, they are also able to share information between related cliques and communities. This means each embedding is able to capture information over a larger total neighborhood: it receives messages not just from it's $K$-hop neighbors in the original graphs, but also from the $K$-hop neighbors of its ancestors in the coarsened graph. As we saw in Table 2, this leads to the ability to influence nodes that are further away in the graph.

To demonstrate the impact of this additional information, we consider a synthetic chain dataset proposed by Gu et al. (2020) to explore ability to capture relationships occurring at longer lengthscales. Each dataset contains 600 chain graphs of length $L$, with labels equally split across two classes. Within each chain, all nodes share the same label. All $100d$ features are non-informative, with the exception of the last node in each chain, whose first two dimensions encode the true label. This means that most nodes do not have any informative nodes in their immediate neighborhood. As we increase the chain length $L$, we therefore increase the importance of longer lengthscales, by increasing the average distance to an informative node feature.

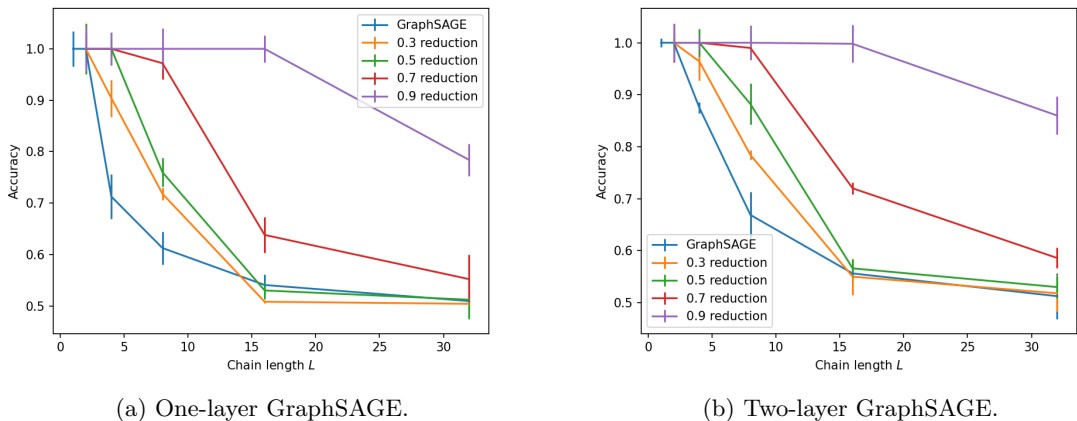

(a) One-layer GraphSAGE.  (b) Two-layer GraphSAGE.

Figure 4: Accuracy on a synthetic chain dataset, with various chain lengths and reduction ratios.

Figure 4a shows how accuracy varies with chain length $L$, applying SMGRL with a single-layer GraphSAGE architecture and varying reduction ratios.[5] For each dataset, we use 20 nodes for training, 100 nodes for validation, and 100 nodes for test. For a chain length of 2, single-layer GraphSAGE is able to achieve 100% accuracy, since all nodes include an informative node in their one-hop neighborhood. As we increase the chain length, the average performance of GraphSAGE decreases, as increasingly many nodes lack information in their one-hop neighborhood. However, SMGRL is able to capture longer-ranging dependencies. With a reduction ratio of 0.5, all length-4 chains are compressed to a length-2 chain in the coarsest graph, and in

---

[5]GraphSAGE on the full graph corresponds to a reduction ratio of 0.

this representation all nodes contain relevant information in their one-hop neighborhood. As we increase the reduction ratio further, the top-level nodes are increasingly likely to include relevant information in their one-hop neighborhood, leading to increased accuracy. We see a similar pattern in Figure 4b, where the underlying architecture is a two-layer GraphSAGE. Here, the baseline algorithm can propagate information from the two-hop neighborhood, and therefore obtains better accuracies than the one-layer counterpart. The general pattern seen in the one-layer case is repeated, however: as the average relevant lengthscale increases beyond 2, we see improved performance.

Table 3: Macro $F_1$ score on Cora datase, with a reduction ratio of 0.4 and various embedding dimensions and aggregation methods. Top: using a single one-layer GraphSAGE model learned on the coarsest level. Bottom: using a separate one-layer GraphSAGE model learned at each level. Stated dimensions correspond to per-level embeddings; concatenated embeddings are twice this. We consider three aggregation methods, plus the embeddings obtained on the original graph (equivalent to the coarsened GCN model of Huang et al. (2021)). For each embedding type, we bold the better of the two results across the two models. Note that the stated embedding dimension is that of the per-level embeddings; the concatenated embeddings are twice this length.

|  | Aggregation | Embedding dimension | | | | | | | |
|  | method | 1 | 2 | 4 | 8 | 16 | 32 | 64 | 128 |
| --- | --- | --- | --- | --- | --- | --- | --- | --- | --- |
| Single GCN | Simple mean | **0.184** | **0.392** | **0.621** | **0.731** | **0.755** | **0.761** | **0.765** | **0.761** |
| trained on | Weighted mean | 0.181 | 0.404 | **0.633** | **0.736** | 0.757 | 0.762 | **0.769** | **0.768** |
| coarsest level | Concatenation | 0.22 | 0.398 | 0.615 | 0.728 | **0.755** | **0.759** | **0.757** | **0.749** |
| Separate GCN/level | Simple mean | 0.18 | 0.372 | 0.574 | 0.695 | 0.741 | 0.754 | 0.756 | 0.747 |
|  | Weighted mean | **0.201** | **0.419** | 0.628 | 0.73 | **0.758** | **0.763** | 0.765 | 0.765 |
|  | Concatenation | **0.352** | **0.569** | **0.692** | **0.742** | 0.754 | 0.755 | 0.743 | 0.741 |

## 5.4 Learning a single GCN does not sacrifice embedding quality

Clearly, incorporating information obtained at multiple levels of granularity can improve performance over a single level of granularity—whether that level is the original graph, or a single coarsened layer. However, under SMGRL, only one level in the hierarchy has embeddings obtained using a GCN trained on that level. A reasonable question might be whether training one GCN per layer, and combining the resulting embedding, does better than our approach. In Table 3 we do exactly that: on the Cora dataset, we compare the SMGRL framework with a single GCN learned at the top level, with a variant that learns a separate GCN for each level. When using our standard weighting aggregation scheme, or using concatenated embeddings, we perform comparably to this variant, for a significantly lower training cost. When using a simple mean aggregation, we uniformly outperform the separate GCNs. This is because using a single GCN means that the embeddings at the two levels are inherently compatible—they aggregate information in the same manner, so each dimension carries similar information. Conversely, two independently trained GCNs will not lead to compatible embeddings, leading to greater loss of information when taking a simple mean.

## 5.5 SMGRL outperforms alternative hierarchical methods

In Figure 5, we compare test set macro $F_1$ scores against two popular hierarchical graph embedding methods, HARP (Chen et al., 2018) and MILE (Liang et al., 2021), implemented using the authors' original code[6][7] with default settings and the same classification architecture as SMGRL (Figure 11 in Appendix D shows analogous accuracy results). HARP generates embeddings on a coarsened graph using node2vec, and refines them as we descend the hierarchy. MILE generates embeddings on a coarsened graph using NetMF (Qiu et al., 2018), and uses a (globally shared) GCN to refine. Since these approaches incorporate both an initial embedding and a refinement, they have higher runtimes than SMGRL. Despite this, we see that SMGRL outperforms both approaches for most settings.

---

[6]`https://github.com/jiongqian/MILE`
[7]`https://github.com/GTmac/HARP`

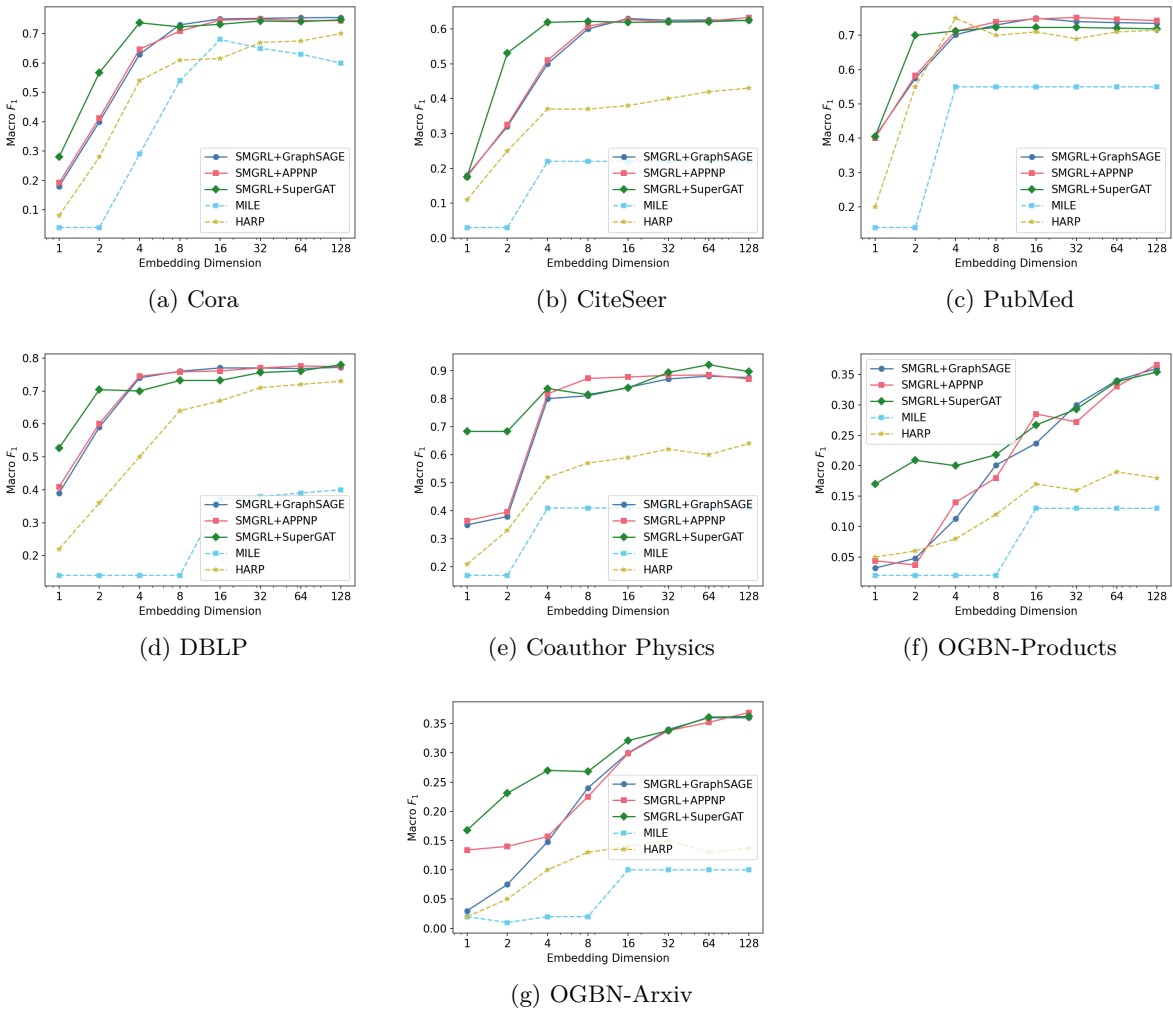

(a) Cora

(b) CiteSeer

(c) PubMed

(d) DBLP

(e) Coauthor Physics

(f) OGBN-Products

(g) OGBN-Arxiv

Figure 5: Macro $F_1$ score on four different graphs, with varying embedding dimensions, for various hierarchical embedding methods. SMGRL uses GraphSAGE with a single-layer architecture, with a reduction ratio of 0.4.

In Appendix D.1, we look at how SMGRL compares with a hierarchical pooling approach, which learns a hierarchy of GCNs coupled via a learned pooling mechanism (Ying et al., 2018). This approach was developed for graph classification; however as we describe in Appendix D.1 it can be adapted to the node classification setting. We find that on small graphs, SMGRL obtains comparable results to the pooling approach, but at a much lower computational cost. This suggests that SMGRL is able to extract relevant hierarchical information in a lightweight manner.

## 5.6  SMGRL can be used in an inductive manner

A key advantage of GCNs over other node embedding methods is that they are inductive. Rather than learn a direct mapping from node to embedding, GCNs learn message passing algorithms that can be applied to arbitrary graphs. Indeed, we directly make use of this when we apply an algorithm trained at one level of the hierarchy, to get embeddings at other levels.

We consider a mixed inductive/transductive setting, where a randomly selected subset of test set nodes are not included in the training graph. We refer to the held-out test set nodes as our inductive test set, and the test set nodes present at training time as our transductive test set. At inference time, we reincorporate the

held-out nodes at each level in our hierarchy. At the bottom level, we simply reintroduce the nodes and any missing edges. At higher levels, we assign the new nodes to the partitions with which they have the highest number of edges. In Figure 6, we show macro $F_1$ on Cora scores for a reduction ratio of 0.4, 8-dimensional embeddings, and a variety of held-out percentages. Error bars show one standard deviation under different random held-out sets. While we do perform better on the transductive test set than the inductive test set, the difference is small relative to the variation across partitions, indicating that SMGRL continues to perform well when a moderate number of new nodes are added to the graph.

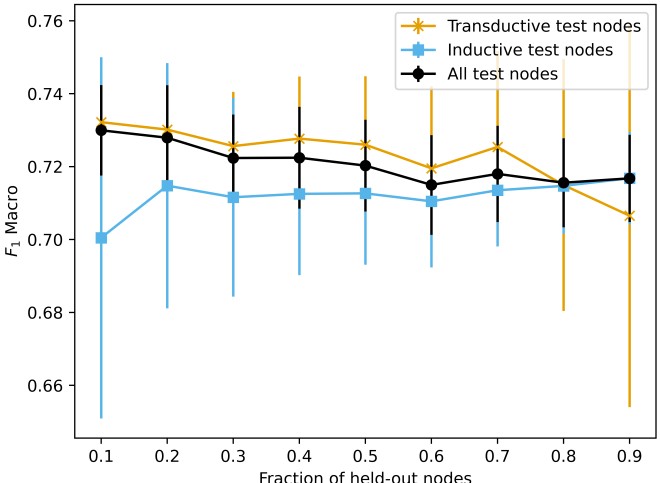

Figure 6: Inductive learning on Cora using SMLRG-GraphSAGE, with a varying percentage of test set nodes held out during training. Macro $F_1$ plotted for overall test set, and for the inductive and transductive subsets.

### 5.7 Distributing inference for large graphs

SMGRL dramatically reduces training time of a GCN, since training is carried out over a smaller graph. However, inferring the final embeddings occurs on the full graph (plus any additional, smaller layers in the hierarchy). While, unlike training, inference only requires a single pass through the graph, this inference cost—and the associated memory requirements—may be infeasible if our graph is large.

In such settings, we propose partitioning the graph at each level of the hierarchy, based on their parents in the graph, to approximate the full graph with a sequence of independent subgraphs. We can then infer the embeddings on each subgraph in parallel, making use of distributed resources if available. This leads to lower computational complexity and memory requirements, but will tend to reduce the quality of the embeddings, since we are ignoring many edges in the original graph. In Figure 7, we compare the macro $F_1$ score of embeddings obtained using the disjoint subgraphs at lower levels, and the embeddings obtained using the full graph, on the Cora dataset. We see that, while there is a drop in performance due to using the disjoint subsets, it is small and may be a worthwhile trade-off in practice.

## 6 Discussion

In this work we introduce SMGRL, an extremely lightweight framework for learning multi-resolution node embeddings. SMGRL is highly customizable and can be applied on top of any GCN. Moreover, its computational cost is approximately equivalent to that of the recently proposed, coarsening-based scalable GCN framework of Huang et al. (2021), while introducing increased expressive power due to incorporating information at multiple lengthscales. Our experiments on real-world datasets show impressive empirical performance. A

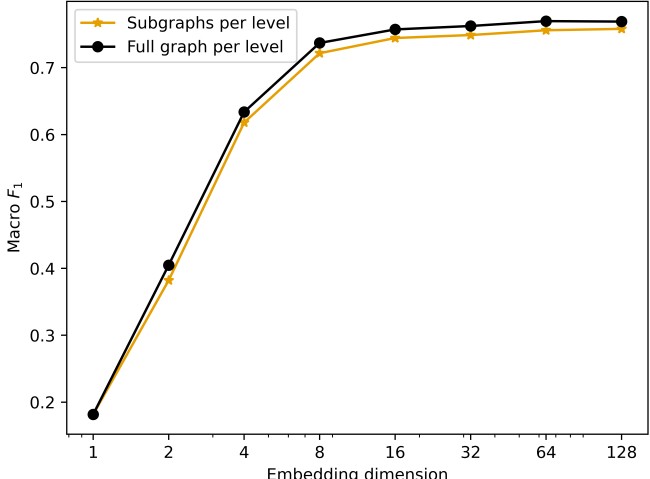

Figure 7: Cora macro $F_1$ score, using full SMGRL-GraphSAGE, and a variant where inference is carried out on a partitioned graph.

future direction might be to incorporate pre-existing partitioning information, such as partitioning users of a social network by employer or country.

As discussed in Section A, SMGRL is primarily applicable on homophilic graphs, since the coarsening algorithms used ignore node labels and so combine nodes based only on structural similarity. While beyond the scope of this paper, the development of graph coarsening algorithms appropriate for heterophilic graphs is an interesting avenue for future research. In addition, we have focused on the task of node classification; we note that GCNs can also be used for graph classification and a number of hierarchical pooling algorithms have been proposed in this setting (e.g., Ying et al., 2018; Xin et al., 2021; Huang et al., 2019; Bandyopadhyay et al., 2020; Liu et al., 2021). We leave exploration of graph classification for future work.

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

## A    Discussion on the coarsening algorithm and applicability of SMGRL

We have chosen to use a coarsening algorithm that retains spectral similarity, since spectrally coarsened graphs have been proven to be good proxies for training GCNs (Huang et al., 2021). However, we ideally *also* want our coarsened graphs to capture meaningful structure within the graph, allowing us to capture relationships between more distant nodes. This sort of representation is what allows hierarchical GCNs such as Guo et al. (2021) and Jiang et al. (2020) to learn rich, multi-scale node embeddings.

To assess whether spectral graph coarsening yields an intuitive hierarchy, we applied spectral coarsening to a number of small, structured graphs: A stochastic blockmodel with 60 nodes (split into clusters of size 10, 20, 30) with edge probability matrix $0.03 + 0.77\mathbf{I}$ (Holland et al., 1983); a circular ladder graph with 50 nodes; a Newman-Watts-Strogatz graph with 100 nodes (Newman et al., 2001), initial clique size of 5, and edge addition probability of 0.1; Zachary's karate club graph (Zachary, 1977); the character co-occurrence graph from *Les Misérables* (Knuth, 1993; Hugo, 1863); and an Erdős-Rényi $G(n,p)$ graph with 100 nodes and edge probability 0.2 (Gilbert, 1959). All graphs were generated using `networkx` (Hagberg et al., 2008).

In Figure 2 we show the hierarchies obtained on each dataset. The left hand side of each subfigure shows the original graph (visualized using `networkx` with a spring layout). Moving towards the right of each subfigure, we see increasingly coarsened graphs. The bottom row shows the coarsened graph; the top row shows the mapping between nodes in the original graph, and nodes in the coarsened graph.

We see that the coarsening algorithm tends to group nodes that are close in the graph, and merge clusters into a single node. For example, in the stochastic block model graph, the coarsest graph contains one node for each latent community (Figure 2a). In the circular ladder graph, the graph is coarsened into a circle, with nodes representing contiguous regions of the original graph (Figure 2b). This suggests that the coarser graphs in the hierarchy do contain relevant information about the graph structure.

This figure also gives us a clue as to what sort of graphs might be well-suited to SMGRL. Figure 2a shows a stochastic blockmodel, which exhibits three clear clusters and no meaningful structure within clusters. We see that the coarsened graphs preserve the clustering structure, with $\mathcal{G}_2$ providing one node per original cluster. We see similar behavior in the Karate club social network (Figure 2d) and the Les Misérables co-occurrence graph (Figure 2e), which also show clear clustering structure. In the circular ladder graph (Figure 2b), we do not have clusters of nodes, but we do have a clear sense of distance between nodes, indicated by position along the ladder. Here, the coarsened graphs compress this distance, splitting the nodes based on position within the circle. We see similar behavior in the Newman-Watts-Strogatz graph (Figure 2c), which features several "chains" of nodes.

By contrast, the Erdős-Rényi graph (Figure 2f) exhibits neither clustering structure nor chains of nodes. The lack of structure means there is no "natural" clustering of the graph, and as a result the coarsened graphs do not capture meaningful information. In addition, some graphs cannot be meaningfully reduced via our approach. In particular, a star graph (where one central node is connected to all other nodes, but no other edges are present) cannot be meaningfully condensed.

These observations suggest that SMGRL will work well on graphs which have some clustering structure, or where there is a significant variation in the distance between two nodes. It will not be appropriate in graphs with no or little structure, since we should not expect coarsening to capture any meaningful information about the graph.

In Table 4, we explore the impact of different choices of clustering algorithm. We look at the performance and total computational time of four coarsening algorithms: The "variation neighborhood" algorithm used in this paper; plus three alternative algorithms also proposed by Loukas (2019) that minimizes the same cost. We see that the four algorithms are comparable in terms of both performance and time.

## B    Computational complexity of SMGRL

We discuss the cost of each step of SMGRL below, noting that the first two steps are identical to the coarsened GCN framework of Huang et al. (2021)

Table 4: Test set $F_1$ score, and total wall time (in seconds), on four datasets, using four clustering algorithms, using two-layer GraphSAGE with a reduction ratio of 0.5.

| Dataset | Coarsening method | Huang et al | | SMGRL | |
|---|---|---|---|---|---|
| | | $F_1$ | Time | $F_1$ | Time |
| Cora | Variation Neighborhoods | $72.4 \pm 0.3$ | 1.11 | $74.4 \pm 0.5$ | 1.25 |
| | Variation Edges | $69.9 \pm 0.7$ | 1.83 | $74.0 \pm 0.3$ | 1.84 |
| | Algebraic JC | $74.3 \pm 0.6$ | 1.61 | $78.8 \pm 0.9$ | 1.89 |
| | Affinity GS | $71.6 \pm 0.1$ | 1.5 | $77.2 \pm 0.6$ | 1.67 |
| DBLP | Variation Neighborhoods | $83.23 \pm 0.6$ | 5.53 | $84.07 \pm 0.2$ | 5.93 |
| | Variation Edges | $83.06 \pm 0.2$ | 5.61 | $84.68 \pm 0.5$ | 5.87 |
| | Algebraic JC | $83.47 \pm 0.4$ | 5.95 | $84.14 \pm 0.2$ | 6.05 |
| | Affinity GS | $83.32 \pm 0.2$ | 5.85 | $84.5 \pm 0.5$ | 5.91 |
| Citeseer | Variation Neighborhoods | $48.8 \pm 0.8$ | 1.61 | $50.6 \pm 0.2$ | 1.38 |
| | Variation Edges | $51.4 \pm 1.3$ | 1.50 | $54.7 \pm 1.2$ | 1.63 |
| | Algebraic JC | $51.3 \pm 0.4$ | 1.32 | $53.3 \pm 0.9$ | 1.61 |
| | Affinity GS | $50.2 \pm 0.7$ | 1.29 | $52.7 \pm 1.4$ | 1.58 |
| Pubmed | Variation Neighborhoods | $68.7 \pm 1.4$ | 3.19 | $70.8 \pm 0.5$ | 3.52 |
| | Variation Edges | $70.4 \pm 0.3$ | 3.60 | $69.0 \pm 0.2$ | 4.04 |
| | Algebraic JC | $73.1 \pm 1.2$ | 4.0 | $72.5 \pm 0.4$ | 4.32 |
| | Affinity GS | $69.5 \pm 0.9$ | 3.88 | $70.3 \pm 1.3$ | 3.99 |

---

**Algorithm 2** The "variation neighborhoods" graph coarsening algorithm (Loukas, 2019)

---

**Input:** Combinatorial Laplacian $L_{\ell-1}$, threshold $\sigma'$, target size $n$.
For each node $v_i$, construct candidate set $\mathcal{C}_i = v_i \cap Ne(v_i)$
$N_\ell = |\mathcal{V}_{\ell-1}|, \text{marked} \leftarrow \varnothing, \sigma_\ell^2 \leftarrow 0$
**while** $|\mathcal{F}_\ell| > 0$ and $N_\ell > n$ and $\sigma^2 \leq \sigma'$ **do**
  $\mathcal{C}^* = \arg\min_{\mathcal{C} \in \mathcal{F}_\ell} \text{cost}(\mathcal{C})$
  $s = \text{cost}(\mathcal{C}^*)$
  $\mathcal{F}_\ell \leftarrow \mathcal{F}_\ell \setminus \mathcal{C}^*$
  **if** all vertices of $\mathcal{C}^*$ are not marked, and $\sigma' \geq \sqrt{\sigma_\ell^2 + (|\mathcal{C}^*| - 1)s}$ **then**
    $\text{marked} \leftarrow \text{marked} \cup \mathcal{C}^*$
    $\mathcal{P}_\ell \leftarrow \mathcal{P}_\ell \cup \mathcal{C}^*$
    $N_\ell \leftarrow N_\ell - |\mathcal{C}^*| + 1$
    $\sigma_\ell^2 \leftarrow \sigma_\ell^2 + (|\mathcal{C}^*| - 1)s$
  **else**
    $\mathcal{C}' \leftarrow \mathcal{C}^* \setminus \text{marked}$
    **if** $|\mathcal{C}'| > 1$ **then**
      $\mathcal{F} \leftarrow \mathcal{F} \cup \mathcal{C}'$
    **end if**
  **end if**
**end while**
Construct $P_\ell$ from $\mathcal{P}_\ell$
$L_\ell \leftarrow P_\ell^{\mp} L_{\ell-1} P_\ell^{+}$
Construct $\mathcal{G}_L$ from $L_\ell$
**Return:** $G_\ell$, $P_\ell$

---

- Constructing a coarsened graph using the neighborhood-based coarsening algorithm of Loukas (2019) scales (up to polylog terms) linearly with both the number of edges, and the product of the number of nodes and the average degree – both of which are upper bounded by $|\mathcal{V}|^2$.

- Training a GCN on the coarsest graph has computational complexity $O(K|\mathcal{V}_L| + K|\mathcal{E}_L|)$ (Blakely et al.), where $K$ is the number of layers in the GCN. Using a coarsening method with reduction ratio $r$, the coarsened graph $\mathcal{G}_L$ has $|\mathcal{V}_L| = (1 - r)|\mathcal{V}|$ vertices. The number of edges $|\mathcal{E}_L|$ in $\mathcal{G}_L$ is upper bounded by $\min\left(|\mathcal{E}|, (1 - r)^2|\mathcal{V}|^2\right)$. If $\mathcal{G}$ is dense, then $|\mathcal{E}| \sim O(|\mathcal{V}|^2)$, suggesting a $(1 - r)^2$ speed-up over the original GCN. If $\mathcal{G}$ is sparse then $|\mathcal{E}| \sim O(|\mathcal{V}|)$, suggesting a $(1 - r)$ speed-up over the original GCN.

- Since the coarsened graphs are smaller than the original graph, for both SMGRL and the coarsened algorithm the inference step is dominated by the cost of inference on the original graph, which scales as $O\left(K|\mathcal{V}| + K|\mathcal{E}|\right)$. Since our hierarchy allows us to extract information at multiple resolutions by design, we choose to let the number of layers, $K$, to be one. In practice the number of iterations required for convergence of the GCN means that in the graphs considered in this paper, the cost of inference is smaller than the cost of training. If desired, inference can be approximated by partitioning the graphs at levels $\ell < L$ based on their parents in level $\ell + 1$, and independently inferring embeddings on the resulting subgraph. This has the additional advantage of allowing parallelization across subgraphs. We empirically explore the resulting performance in Section 5.7.

- Aggregating the embeddings and learning the final classification algorithm is linear in the number of labeled vertices $|\mathcal{V}|$, which is typically smaller than $|\mathcal{E}|$.

## C    Additional experiments comparing SMGRL with single-level embeddings

In Section 5.2, we looked at how using SMGRL improves performance over both the underlying GCN, and the single-layer coarsening-based approximation of Huang et al. (2021). In Figure 8, we repeat this analysis using two-layer GraphSAGE, APPNP and SuperGAT. In general, we see better performance with an additional layer (albeit at a higher computational cost). In particular, the performance of GCNs trained on the full graph has improved over Figure 3, due to the ability to capture relationships occurring at longer lengthscales. As a result, SMGRL is no longer the clear winner in terms of performance, with the unmodified GCNs performing better in many (but not all) cases. However, the faster SMLRG method remains competitive with the full model, and as before, performs better than single-layer embeddings.

In Section 5.2 and Figure 8, we look at how macro $F_1$ scores vary for a fixed reduction ratio and varying embedding dimensions. In Figure 9, we show equivalent plots holding the embedding dimension fixed at 8 and with varying reduction ratios. Note that the number of levels of the hierarchy are determined by the reduction ratio. As before, the level 0 embeddings are equivalent to the approach of Huang et al. (2021). We see a similar pattern: For all but the highest reduction ratio, the GCN learned on the coarsest graph performs well at all layers, typically leading to embeddings that perform better than those obtained using a GCN trained on the original graph (indicated by a reduction ratio of zero). And in almost all cases, an aggregated embedding which considers all levels of the hierarchy outperforms any single level's embeddings.

In most of our experiments, we have reported the macro $F_1$ score, since it is more informative than accuracy when dealing with unbalanced classes. However, we see similar trends when looking at accuracy. To demonstrate this, in Figure 10, we show the accuracies obtained using SMGRL aggregated embeddings and per-level embeddings, over various embedding dimensions (i.e., the same scenarios for which we report macro $F_1$ in Figure 3). As is the case for macro $F_1$ score, we that SMGRL obtains better accuracies than the baseline GCN and the approach of Huang et al. (2021) (corresponding to Level 0 embeddings) when using GraphSAGE and APPNP. As before, we do not strictly outperform the comparison methods when using SuperGAT on PubMed, DBLP and Coauthor Physics; however we achieve comparable accuracies.

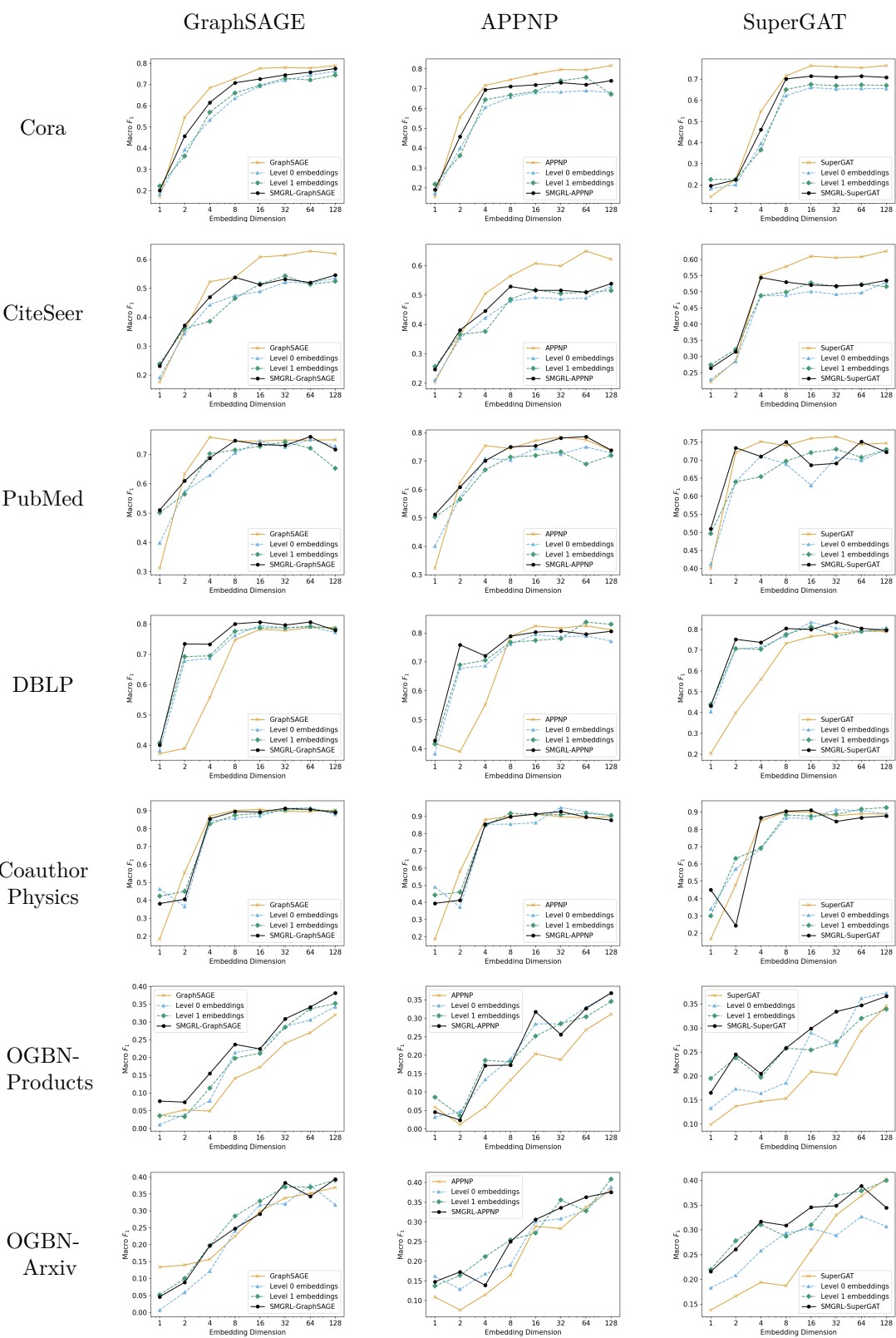

Figure 8: Test-set macro $F_1$ score for SMLRG embeddings obtained at different levels of the hierarchy, using three two-layer GCN architectures (GraphSAGE, APPNP, SuperGAT), a reduction ratio of 0.4 (resulting in a two-layer hierarchy), and various embedding dimensions. Note, the results for Level 0 embeddings directly correspond to the method of Huang et al. (2021).

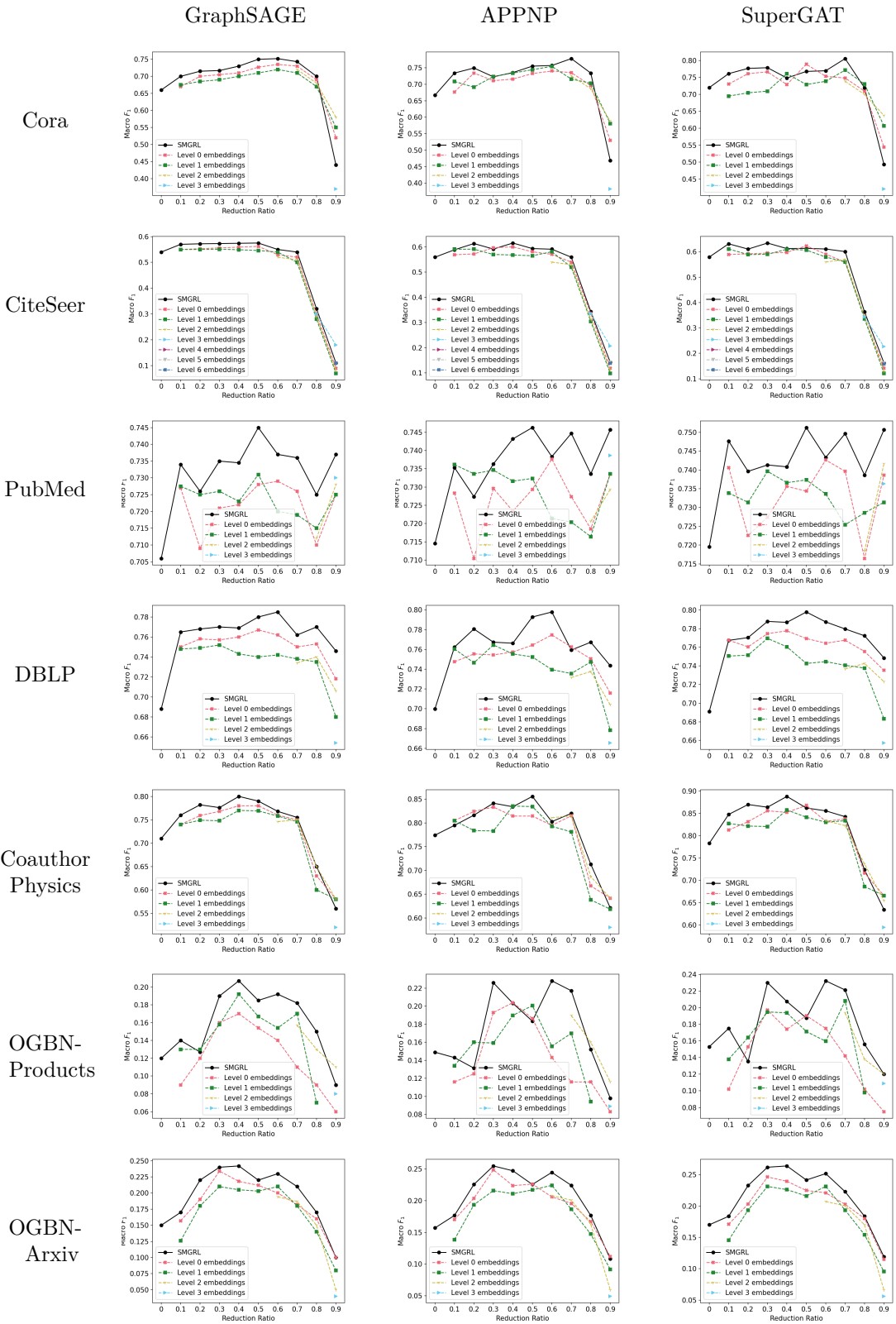

Figure 9: Test-set macro $F_1$ score for SMLRG embeddings obtained at different levels of the hierarchy, using three single-layer GCN architectures (GraphSAGE, APPNP, SuperGAT), an embedding dimension of 8, and various reduction ratios. Note that a reduction ratio of zero corresponds to the GCN on the original graph. Note, the results for Level 0 embeddings directly correspond to the method of Huang et al. (2021).

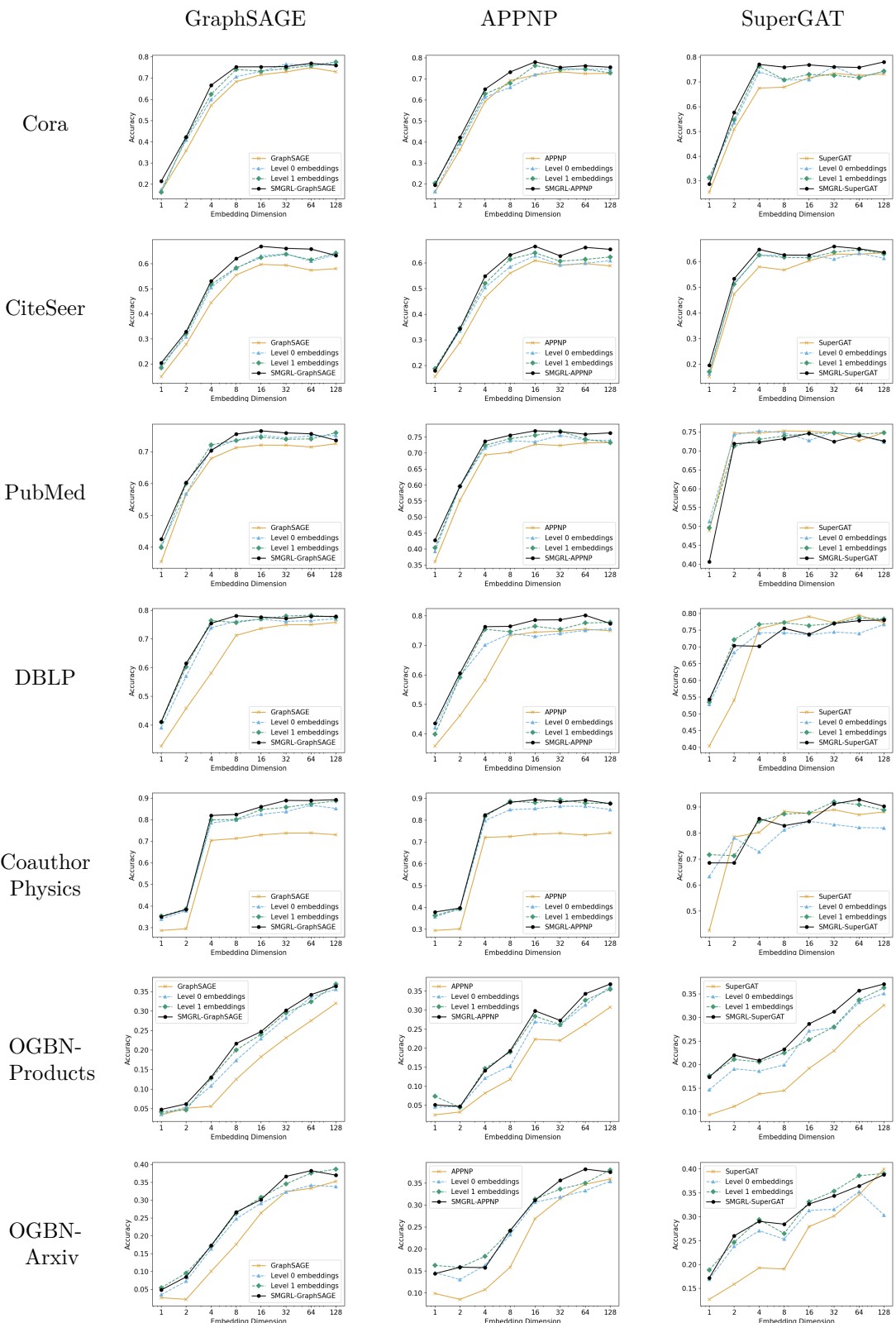

Figure 10: Test-set accuracies for SMLRG embeddings obtained at different levels of the hierarchy, using three single-layer GCN architectures (GraphSAGE, APPNP, SuperGAT), a reduction ratio of 0.4 (resulting in a two-layer hierarchy), and various embedding dimensions. Note, the results for Level 0 embeddings directly correspond to the method of Huang et al. (2021).

Table 5: Test set $F_1$ score and total training time for DiffPool (modified for node classification task, see Appendix D.1) and SMGRL, on two datasets, with a two-layer hierarchy with reduction ratio of 0.4.

| | Embedding Dimension | DiffPool $F_1$ score | Time (s) | SMGRL $F_1$ score | Time (s) |
|---|---|---|---|---|---|
| Cora | 8 | 0.72 | 79.04 | 0.73 | 1.38 |
| | 16 | 0.73 | 80.01 | 0.75 | 1.52 |
| Citeseer | 8 | 0.54 | 81.02 | 0.57 | 1.47 |
| | 6 | 0.58 | 81.46 | 0.58 | 1.58 |

## D  Additional experiments comparing SMGRL with alternative hierarchical methods

### D.1  Comparison with hierarchical pooling methods

As we discussed in Section 2.2, hierarchical pooling methods have been proposed to learn richer graph embeddings. Such methods construct a hierarchy of increasingly coarsened graphs, and jointly learns GCNs at each layer of the hierarchy. The layers are coupled, such that the input node features in $\mathcal{G}_\ell$ are derived from the learned embeddings from $\mathcal{G}_{\ell-1}$ via a pooling operation. This pooling operation is itself learned—for example using a GCN (Ying et al., 2018) or an attention mechanism (Huang et al., 2019).

While these methods have been developed for learning graph-level embeddings, they exhibit clear similarities with SMGRL, albeit with higher computational cost (since we are learning multiple GCNs—including one on the full graph—plus pooling operations in an end-to-end manner, rather than learning a single GCN on a coarsened graph). Here, we explore whether this end-to-end hierarchical approach leads to improved performance over SMGLR, in a node classification setting.

To measure this, we modify the DiffPool algorithm (Ying et al., 2018) to accommodate a node classification task. Rather than simply using the final pooled graph embedding, we combine the per-level node embeddings in a manner analogous to SMGLR: we lift the coarse embeddings back to the original graph using the learned coarsening matrix, and combine the per-layer embeddings via a mean operation.

In Table 5, we compare the performance of this modified DiffPool algorithm with SMGRL. We use a single layer GraphSAGE algorithm as our underlying GCN, and both methods use a two-layer hierarchy with a reduction ratio of 0.4. Note, due to the increased computational and memory requirements of DiffPool, we have only considered the smallest graphs for this analysis (DiffPool jointly learns one GCN per layer of the hierarchy, plus pooling layers, and is not designed as a scalable algorithm). We see that SMGRL performs comparably with DiffPool in this setting, but for a much lower computational cost.

### D.2  Additional comparisons with MILE and HARP

In Section 5.5, we compared SMGRL with MILE and HARP, looking at macro $F_1$ score. In Figure 11, we repeat this analysis, this time looking at accuracy. As before, we see that SMGRL outperforms both alternative hierarchical methods for most settings.

## E  Exploration of alternative final aggregation schemes

As we have seen in the experiments so far, learning post-hoc weights to combine the embeddings works well in practice. However, this does add additional computational complexity (albeit minimally, as described in Appendix B), and limits our choice of classifier to one that can be jointly trained with the post-hoc weightings. In Figure 12, we compare our post-hoc weighting scheme with two alternatives: taking a simple mean, and directly concatenating the embedding. All experiments use single-layer GraphSAGE, with a reducion ratio of 0.4 and varying embedding dimensions. Note that embedding dimension here refers to the per-layer embedding dimension; the concatenated dimension will be multiplied by the number of dimensions in the hierarchy. We see that in general, the learned weighted mean performs better than the alternative approaches;

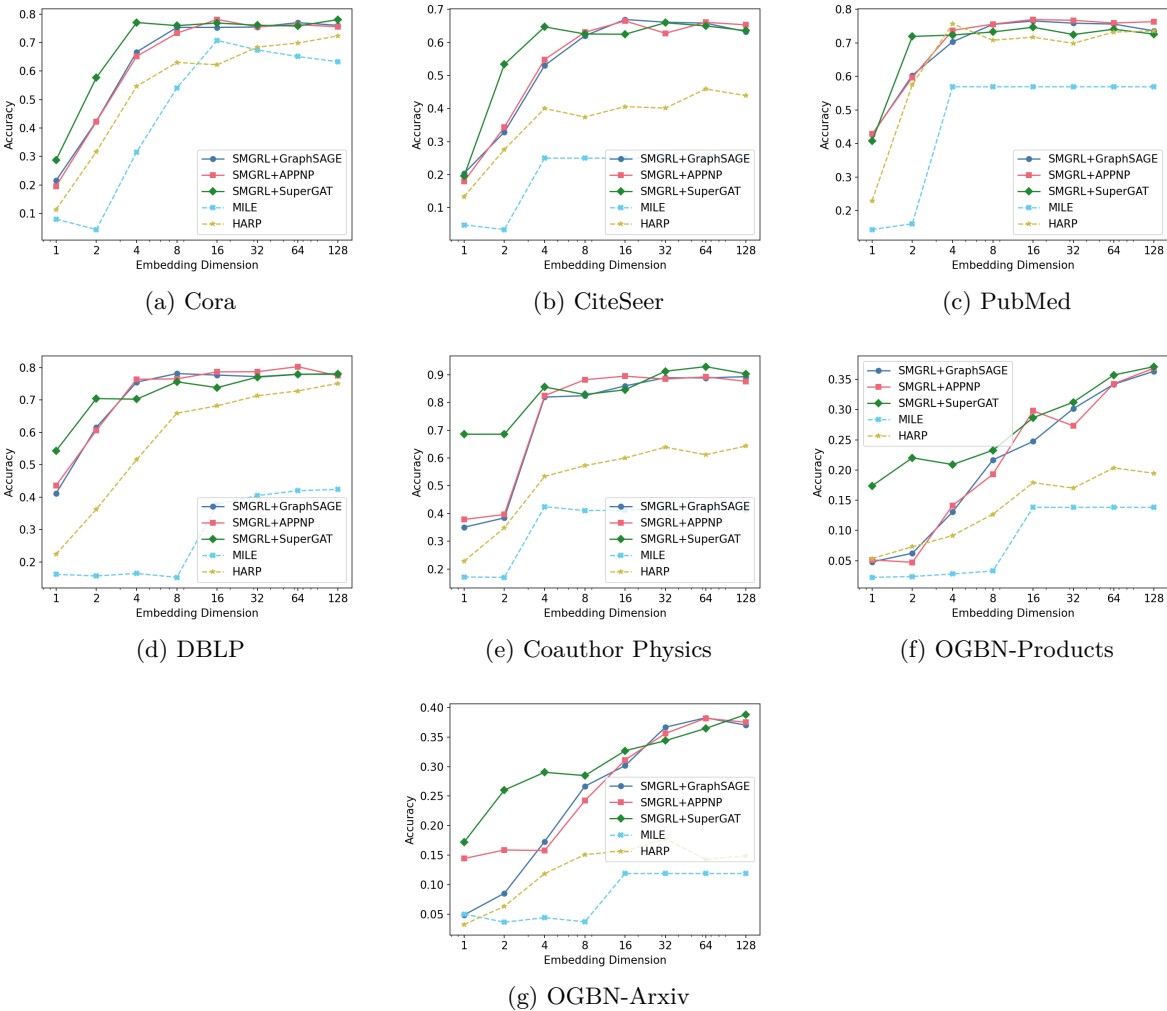

Figure 11: Accuracy on four different graphs, with varying embedding dimensions, for various hierarchical embedding methods. SMGRL uses GraphSAGE with a single-layer architecture, with a reduction ratio of 0.4.

however the difference is fairly small. This leads us to suggest using a simple mean if computational resources are limited.

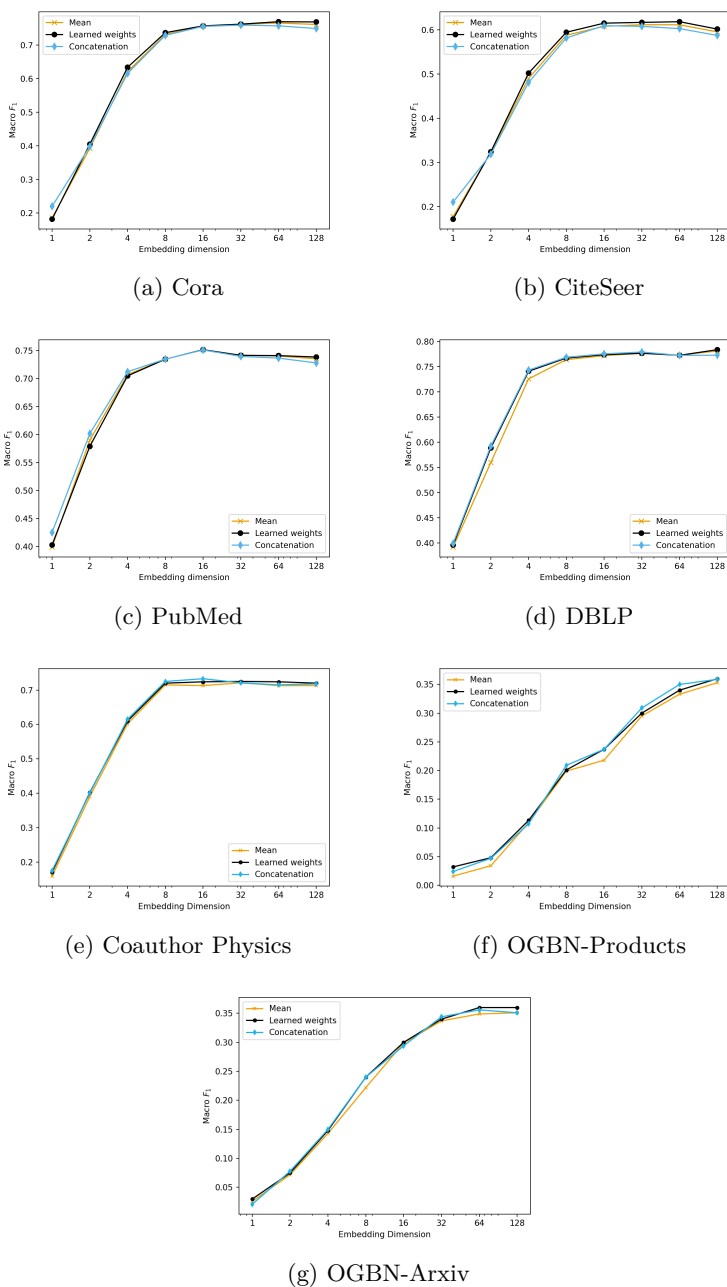

Figure 12: Macro $F_1$ score on seven different graphs, for three different aggregation schemes, using SMGRL + single-layer GraphSAGE. The reduction ratio is fixed at 0.4, and a variety of embedding dimensions are shown.

