# OpenReview forum: "SMGRL: Scalable Multi-resolution Graph Representation Learning"
_TMLR — Rejected by TMLR_

### Review · Reviewer_muSz · 2023-09-02

**Summary Of Contributions:**

To obtain the node embeddings of graphs with both short-range and long-range dependencies, the paper proposes a framework named SMGRL that gains the hierarchy of a graph by the coarsening operation. Based on this, a GCN is trained with the correct and smallest subgraph for reducing training time, while the inference is also based on the afore-obtained graph hierarchy and trained GCN. The empirical results on several datasets show that the proposed SMGRL is competitive.

**Audience:**

Yes

**Claims And Evidence:**

Yes

**Requested Changes:**

Please refer to the above Strengths and Weaknesses.

**Strengths And Weaknesses:**

Pros

- The investigated problem is novel and important.
- The paper is easy to follow.
- The empirical results of SMGRL are good and competitive.
- Some visualizations of the graph hierarchy are provided.
- The source code is provided for reproduction.

Cons

- The concept of short-range and long-range dependencies is not clearly defined.
- Basically, the paper is more like a technical report that designs a new framework. The paper is full of technical details with only marginal novelty.
- However, the key technical part of the paper, i.e., graph coarsening, is not clearly elaborated. It seems that the paper directly utilizes an existing approach for coarsening.
- Although some small/medium-scale OGB datasets are considered, their scale is too small and, hence, insufficient to justify the efficiency and scalability of the proposed framework.
- I would suggest involving more large-scale datasets and providing a thorough efficiency comparison with baselines.
- Besides, the long-range graph benchmarks should also be considered for experiments, which is essential.
- There is no in-depth analysis of the researched problem nor any kind of theoretical analysis.
- The writing and presentation should be largely improved.
- The paper is redundantly written but does not highlight the key points, and the paper's core novelty is difficult to reflect.

---

> ### Author Response · Authors · 2023-10-12
>
> Thank you for your comments and suggestions. We have edited the paper to improve clarity, based on your suggestions and those of the other reviewers. We answer specific questions below:
>
> **Long and short-range dependencies**
>
> We have added the following sentence to the beginning of Section 3: “Often, the label of a node in a graph is related not only to the labels of its immediate neighbors and the structure of its immediate neighborhood (properties we refer to as short-range dependencies) but also to the labels of nodes further away in the graph and larger-scale structural properties of the graph (properties we refer to as long-range dependencies).”
>
> We have also added details to the first use of the phrase “long-range dependencies” in the introduction: “A single-layer GCN aggregates information only from a node's immediate neighborhood, ignoring any longer-range dependencies (i.e, relationships between nodes that are further apart in network distance than is captured by the immediate neighborhood)”
>
> **Details of the coarsening approach**
>
> We do indeed make use of existing coarsening approaches and have added more clarification of these approaches both in Section 3 and in the appendix. Our novelty is in using the clustering approach as the basis of a multi-resolution embedding. Previous work (Huang et al, 2021) has used similar clustering methods to learn a single-resolution embedding, by applying a GCN trained on a coarsened graph to the original graph. We show that combining representations obtained using both coarsened and original graphs improves performance with a minimal increase in computational and memory cost over the approach of Huang et al (2021). Further, we dramatically increase scalability while typically improving performance when compared with the base GCN algorithm trained on the original graph.
>
> **Scale of graphs**
>
> We include larger graphs than are considered in the closest related work (Huang et al, 2021). All graphs included in that work are also included here. While our approach increases scalability, we still need to run competing algorithms to assess relative performance; unfortunately, we have limited access to computational resources which restricts the size of the graph we can consider.
>
> **Efficiency compared with baselines**
>
> We have added timing results in the appendix, showing that the computational cost of our method is comparable with that of Huang et al (2021). Please note, that the current set of numbers reflects experiments carried out on a laptop—competing resources for cluster computing meant we were not able to carry out experiments on better hardware. As a result, the numbers reported for Huang et al’s method are much slower than those described in that paper. We will update these numbers once we are able to obtain results using better hardware.

---

### Review · Reviewer_GXyM · 2023-09-26

**Summary Of Contributions:**

In graph-structured data, it is common to seek vector representations for nodes, known as node embeddings. These embeddings, which capture the topological structure of a graph, can be utilized for tasks such as node classification and link prediction. The prevalent technique for this is the use of Graph Convolutional Networks (GCNs), which learns how to obtain these embeddings by considering neighbors of a node’s. GCNs stand out due to their inductive ability, allowing them to work on new or modified graphs, unlike transductive algorithms that cannot generalize beyond the training graph. However, GCNs have limitations, such as only considering immediate neighbors, which can lead to oversmoothing when adding more layers. Moreover, they come with computational and memory challenges, especially as the size of the graph scales.

The proposed solution to alleviate computational and memory demands is to train the GCN on a "similar" smaller graph, using coarsening algorithms. However, this does not entirely address the challenge of capturing long-range dependencies in the graph without adding more layers. To combat this, a hierarchy of graphs can be created, where nodes are aggregated into "supernodes" at subsequent levels. This hierarchical approach captures both long-range and local interactions, allowing for the inference of multi-resolution embeddings for each node without the need for extensive computations at each hierarchical level. This proposed algorithm, called Scalable, Multi-resolution Graph Representation Learning (SMGRL), has been found to be more effective than GCNs trained on the original or coarsened graphs and outperforms other hierarchical GCN algorithms while being more computationally efficient. SMGRL comprises three main components: hierarchical graph coarsening, learning a GCN on this coarsened graph, and a final aggregation of the results.

**Audience:**

Yes

**Claims And Evidence:**

Yes

**Requested Changes:**

1. Please add results for heterophilic datasets.
2. Please compare with pooling methods, to just demonstrate the various trade-offs involved.
3. Please add an ablation study with various coarsening methods.
4. Please add memory usage and timing comparisons on various datasets.
5. Please add pseudocode for coarsening algorithm in the appendix, and add the projection matrices as output of the COARSE in Algorithm 1 to clarify where they are obtained.

**Strengths And Weaknesses:**

# Strengths
1. **Idea**: The key innovation in the paper is learning GCNs at multiple level and combining the information for the final task.
2. **Results**: While the model suffers somewhat in the lower sized graphs, but it seems to do well for large sized graphs.
# Weaknesses
1. **Missing Baselines**: Albeit, this method approaches it differently, there is similarity between the proposed approach and the hierarchical pooling methods, so it would be good to compare against them as well, even if to just demonstrate the tradeoffs involved. Unfortunately, none of the prior works MILE and HARP have compared against pooling methods either.
2. **Missing Datasets**: The long range interaction behavior via synthetic data. However, it is known in graph literature that long range interaction is important for heterophilic datasets [A] These datasets would have formed an excellent way to demonstrate the long range capturing aspect of the proposed model.
3. **Effect of Coarsening**: Since approaches reliant on coarsening largely depend on the quality of coarsening, Huang et. al, demonstrated various coarsening algorithms and how their approach performed with the different coarsening algorithms. A similar analysis here would have been useful.
4. **Timing and Memory Comparison**: The paper claims speed up in many places in the paper, however, there is just one section that provides the complexity analysis. It would be good to actually see the timing comparisons on various datasets and also memory usage.
5. **Clarity**: The coarsening method used here is directly taken from the paper of Loukas et. al. However, it would be good to reproduce a pseudocode of the coarsening algorithm in the appendix. Also, in Algorithm 1, in Step 6, $P_1^{+}, P_2^{+}, \ldots$ etc. is used directly without any indication of where they are obtained from. I suspect that these should be the output of $\textrm{COARSE}$.

[A] Adaptive Universal Generalized PageRank Graph Neural Network, ICLR 2021.

---

> ### Author Response · Authors · 2023-10-12
>
> **Comparison against pooling algorithms**
>
> We agree that the approaches in this paper could be applied to graph classification allowing comparison with pooling algorithms; however, this may require some alteration of the aggregation methodology proposed in the paper. We have chosen to focus on the node classification since this is the primary focus of the most closely related works, MILE, HARP and Huang et al (2021). We have edited Section 6 to reflect this.
>
> **Heterophilic datasets**
>
> Thank you for your suggestion to consider heterophilic graphs.
>
> The coarsening algorithms used in this paper look only at the topology of the graph; they do not take into consideration node labels. This is not a problem in homophilic graphs, where neighbors tend to have similar labels, meaning the nodes in each coarsened graph are likely to correspond to a set of original nodes with the same label. However, in heterophilic graphs, nodes in a coarsened graph are likely to combine nodes with different labels in the original graph. As a result, the labels of the coarsened graphs will become increasingly non-discriminative, limiting our ability to learn. While beyond the scope of this paper, the development of graph coarsening algorithms appropriate for heterophilic graphs is an interesting avenue for future research.
>
> We have carried out some preliminary experiments on two heterophilic graphs -- Squirrel and Chameleon -- and found that, indeed, our model does not perform well on these graphs, supporting these insights. Further, the algorithm of Huang et al (2021) (which also relies on graph coarsening) does not perform well on these graphs. We have made it clearer in the paper that our focus is on homophilic graphs and added the discussion above about the reasons for this (See Section 3.1).
>
> **Effect of Coarsening/Timing**
>
> Thank you for the suggestion of exploring alternative coarsening algorithms and timing algorithms. In Appendix B, we have included experiments comparing alternative coarsening algorithms, mirroring the table from Huang et al (2021). Please note: While we included timing results, these are based on experiments carried out on a laptop; due to competition for resources due to the ICLR deadline we were not able to run these experiments using better hardware. The resulting timing numbers are therefore significantly greater than those reported in Huang et al (including for direct comparisons, i.e. our implementation of Huang et al (2021)). We will add updated timing results once we have completed the associated experiments.
>
> **Clarity/pseudocode**
>
> We have added more details on the coarsening algorithm in Section 3, plus the pseudo-code of the variation neighborhood algorithm in the Appendix (Algorithm 2), which we now refer to from Algorithm 1.
> We have clarified the notation used in the coarsening algorithm. $P^+$ is the pseudo-inverse of $P$. We have edited Equation 1 to be more concrete.

---

> > ### Comment · Reviewer_GXyM · 2023-10-12
> > **Response to Comment 2023/10/12**
> >
> > **Comparison against pooling algorithms:** The suggestion was not to do graph classification but to compare pooling methods on node classification. There is a very simple way to use models like DiffPool for node classification following your approach. Each diff pool layer outputs an $S^l$ which could be used to construct the labels for that pooled graph. Thereby, you can optimize for the node labels at each layer just like your proposed approach. This strong resemblance in the approach is why the comparison is important, as it would establish the trade-off in using external coarsening methods versus differentiable pooling.
> >
> > **Heterophilic datasets:** This is good to know. Thanks for clarifying this in the writing as well. Additionally, it would be good to avoid terms like _long-range dependencies_ (since the term is associated with heterophilic graph models) with something that better reflects the precise expectation here.
> >
> > **Effect of Coarsening/Timing:** It looks like Variational Neighborhoods may not necessarily be the best coarsening method and different dataset could benefit from different coarsening methods. You may want to rewrite Section 3.a, so that it is not too focused on Variational Neighborhood algorithm and present an abstraction that befits the use of multiple coarsening methods.
> >
> > **Clarity/pseudocode:** Thanks for taking the effort to add more details.

---

> > > ### Author Response · Authors · 2023-11-07
> > >
> > > We appreciate the reviewer’s constructive feedback and helpful suggestions. We have revised our paper accordingly and addressed the reviewer’s comments in detail below.
> > >
> > > **Pooling algorithms:** Thank you for the additional information on pooling algorithms. While the papers for DiffPool and related algorithms only evaluate on a graph classification setting, we agree that they can be modified for a node classification setting. In Appendix D.1 we now describe how such a method can be modified, and provide some results comparing DiffPool with SMGRL. The performance is similar; however the computational cost of SMGRL is much lower, since we only train on a coarsened graph. By contrast, DiffPool is jointly training GCNs at each layer of the hierarchy, plus a pooling operation between layers of the hierarchy.
> > >
> > > **Heterophilic datasets:** We have changed our notation to refer to interactions occurring at various length-scales, rather than long vs short range dependencies. We have added text defining the length-scale of an interaction between two nodes as the minimum path distance between these two nodes. In addition, we have added to Section 5.2 a table showing that, for a single-layer GCN, the average length-scale of interactions obtained using a coarsened graph is between two and three times that of the average length-scale on the original graph.
> > >
> > > **Coarsening methods:** We agree that our results do not depend on the specific “variation neighborhoods” algorithm of Loukas, although other approaches lack the theoretical guarantees of that algorithm. We have rewritten the description of the coarsening step to make this point, de-emphasizing the importance of the specific algorithm and pointing to the appropriate experimental results.

---

### Review · Reviewer_d1at · 2023-09-27

**Summary Of Contributions:**

Proposes a hierarchical method to obtain vertex embeddings in graphs for vertex classification. The method coarsens the graph in multiple granularities, learns a GCN on the coarsest granularity, runs it on all granularities, and finally aggregates the resulting embeddings per vertex. Performs an experimental study on OGB datasets comparing to some baselines.

**Audience:**

Yes

**Claims And Evidence:**

No

**Requested Changes:**

W2 and W3 need to be addressed, which is a substantial change. Afterwards, the paper should be reviewed from scratch.

**Strengths And Weaknesses:**

Although the methods does have its strength, they are outweighed by its weaknesses in the current draft. I'll thus focus mainly on these weaknesses.

Strengths.

S1. Simple method. It's quick to implement and easy use.

S2. Can be used with different message-passing GCNs.

S3. Significant savings in training time.

Weaknesses.

W1. Low novelty. The proposed method is close to prior work on using GCNs in a hierarchical fashion and, in particular, to Huang et al. (2021). Huang et al. (2021) train a model on the coarse graph and apply it to the original graph; this paper also trains a model on the coarse graph, but now applies it to intermediate coarsenings and aggregates. The strengths listed above are all shared with Huang et al. (2021).

W2. Unclear if approach improves over prior work. This is for two main reasons: (1) The paper uses non-standard metrics (macro F1), whereas everyone else in this area reports Accuracy+StdDev. The results thus cannot be compared to published results. (2) The paper compares only to "flat" baselines GCNs and two selected hierarchical methods. Other hierarchical methods (both newer ones, but also Huang 2021) do not appear in the study, as are recent "graph transformer" methods. In addition, hyperparameter tuning is not discussed at all, neither for the proposed not the alternative methods.

W3. No analysis/insight. The paper does not provide substantial arguments for its approach beyond that of prior work; the discussion is largely superficial or "story telling". Key assumptions are not spelled out (undirected graph, homophily, no edge features, vertex classification). Key algorithmic aspects are not discussed (e.g, how to aggregate features/labels along the graphs and how they affect the coarsening). The experimental study provides little insight into why the proposed method works better.

W4. Presentation needs work. I found the presentation lengthy, given the simplicity of the method. Also the presentation of the experimental results is cumbersome: there are many plots, which is wasteful in terms of space and makes comparison of performance across different settings painful.

---

> ### Author Response · Authors · 2023-10-12
>
> Thank you for your review and constructive feedback. We are glad that you appreciate the flexibility of our method and its benefits in terms of computational time.
>
> **Novelty and improvement over prior work**
>
> We agree that our model builds heavily on Huang et al (2021), and have emphasized this in the text.  Our “level 0” results implement the algorithm of Huang et al (2021); we have made this clearer in the text.
>
> While our model shares computational properties with Huang et al (2021), by applying the resulting algorithm at multiple levels of hierarchy, we improve performance over both Huang et al (2021), and over the original GCN (see Figures 3 and 8 (previously 4), plus the newly added Figure 10). We have edited the text in Section 5.2 to emphasize this improvement over Huang et al (2021).
>
>
> **Non-standard metrics**
>
> We opted for the macro F1 score as it is a more informative metric when dealing with imbalanced classes, which is often the case in graph data. However, we have now added some additional accuracy results in the appendix (Figures 10 and 11); these results show similar trends as the F1 score results in the main paper.
>
>
> **Hyperparameter tuning**
>
> Wherever possible, we used parameter values from the original GCN implementations, to ensure a fair comparison. We did explore sensitivity to these parameter settings, particularly the choice of optimizer and learning rate, using the Optuna library(specifically, the Nondominated Sorting Genetic Algorithm II). We found that tuning the hyperparameters had little effect on the final results, and so chose to stick with the default parameters. We have added more detail to Section 5.1
>
> **Implementation details**
>
> We have expanded the discussion of the coarsening step (Step a)  in Section 3, adding more details of the variation neighborhood algorithm and detailing how we propagate node features and labels. We have also included the pseudocode of the variation neighborhood algorithm in the Appendix. We have added details on modeling assumptions in Sections 3, Section 3.1, and Appendix A (which now contains information and plots previously in Section 3.1). In short: our model assumes undirected, unlabeled graphs since these are required by our choice of coarsening algorithm. In addition, we find SMGRL works best on homophilic graphs since the coarsening algorithm ignores graph labels and will tend to yield uninformative coarsened labels on heterophilic graphs; we now discuss this in Section 3.
>
> **Analysis/Why our method works better**
>
> Our experimental results in Section 5.3/Figure 4 (previously Figure 5) indicate why we believe our model outperforms baseline GCNs and the approach of Huang et al (2021): The coarsened graphs contain information about the relationship between nodes that are not connected in the original graph. We see this explicitly in the example in Section 5.3: under a K-layer GCN (or coarsened version as per Huang et al, 2021), information can only propagate between two nodes if they are within K edges of each other. However, these graphs contain information that requires longer range propagation of information, due to the low number of labeled edges. SMGRL propagates information along a lengthscale of K for each coarsened graph; in the coarsest graphs, this corresponds to a distance greater than K. We have expanded our discussion in Section 5.3 to make clearer why this example needs a model that incorporates long-range dependency, and why SMGRL outperforms a baseline GCN that operates only on the original graph.
>
> **Presentation/number of plots**
>
> Thank you for the suggestion to cut down the presentation. Based on this, we have moved most of the previous Section 3.1 (Discussion on the coarsening algorithm and applicability of SMGRL) to the appendix (Appendix A). We have also moved results on 2-layer graphs (previously, Figure 4) to Figure 8 in Appendix B (previously, Appendix A).

---

### Author Response · Authors · 2023-10-12

We thank the reviewers and the action editor for their careful and thoughtful comments on our paper.

Based on the feedback we have revised the document, taking into account clarity concerns brought up by the reviewers. In particular, we have clarified the assumptions made by our algorithm and added a discussion of why SMGRL is not well-suited to heterophilic graphs. We have added additional timing experiments, and experiments exploring alternative choices of coarsening algorithm and comparing timing to Huang et al (2021). While we feel that the F1 score is more reflective of performance in the case of class imbalance, we have added additional accuracy results to the appendix (Figures 10 and 11).

We go into these changes in more detail, and respond to individual questions, in our responses to the individual reviewers.

---

### Decision · Action_Editor_J42c · 2023-11-07

**Recommendation:** Reject

**Comment:**

To obtain the node embeddings of graphs with both short-range and long-range dependencies, the paper proposes a framework named SMGRL that gains the hierarchy of a graph by the coarsening operation. Based on this, a GCN is trained with the correct and smallest subgraph for reducing training time, while the inference is also based on the afore-obtained graph hierarchy and trained GCN. The empirical results on several datasets show that the proposed SMGRL is competitive.

However, the current version still has some problems. For example, 1) Long-range interactions: The phenomena of long-range dependencies as explored in this paper is different from what has been traditionally understood in the heterophilic graph node classification community. Gu et. al., which studies the chains dataset used this dataset to demonstrate the utility for explaining their model behavior on an inductive dataset like PPI. However, the current proposed approach in the paper is only for transductive node classification setting and if the long range dependency is not useful for heterophily, then it is quite unclear as to what purpose it serves in this setup. Not having this clarity, again, reduce the novelty aspects of the paper significantly. 2) Improved Explanation: As noted in the response to authors, the current writing of the coarsening is tightly integrated with Loukas et. al algorithm, however, the ablation has shown that different coarsening algorithms can benefit different dataset. It would be preferred to rewrite this work where any coarsening algorithm can be plugged and used. However, this is a significant rewrite of the paper and thus would require a re-submission of the work. 3) Stronger submission: There are some existing methods like DiffPool that one can modify in line of what the paper is proposing and used. This approach will also coarsen the graph as what the paper is proposing to do, but without using a deterministic coarsening algorithm. Comparison of the proposed algorithm with such modifications could make for a more stronger submission.

Therefore, we cannot accept this work this time, but the authors are encouraged to resubmit after a major and significant revision. We will consider to recommend its acceptance if the authors had addressed these issues properly.

**Audience:**

Yes

**Claims And Evidence:**

The current version still has some problems. For example, 1) Long-range interactions: The phenomena of long-range dependencies as explored in this paper is different from what has been traditionally understood in the heterophilic graph node classification community. Gu et. al., which studies the chains dataset used this dataset to demonstrate the utility for explaining their model behavior on an inductive dataset like PPI. However, the current proposed approach in the paper is only for transductive node classification setting and if the long range dependency is not useful for heterophily, then it is quite unclear as to what purpose it serves in this setup. Not having this clarity, again, reduce the novelty aspects of the paper significantly. 2) Improved Explanation: As noted in the response to authors, the current writing of the coarsening is tightly integrated with Loukas et. al algorithm, however, the ablation has shown that different coarsening algorithms can benefit different dataset. It would be preferred to rewrite this work where any coarsening algorithm can be plugged and used. However, this is a significant rewrite of the paper and thus would require a re-submission of the work. 3) Stronger submission: There are some existing methods like DiffPool that one can modify in line of what the paper is proposing and used. This approach will also coarsen the graph as what the paper is proposing to do, but without using a deterministic coarsening algorithm. Comparison of the proposed algorithm with such modifications could make for a more stronger submission.

Meanwhile, as observed by multiple reviewers, the overall novelty of this work is limited. This work largely builds on top of Huang. et. al. and the key insight here is that utilization of model learning at different levels of coarsening. However, this point will not been counted for the overall decision for TMLR journal.

**Resubmission Of Major Revision:**

The authors may consider submitting a major revision at a later time.